# Plasma Gelsolin Inhibits Natural Killer Cell Function and Confers Chemoresistance in Epithelial Ovarian Cancer

**DOI:** 10.3390/cells13110905

**Published:** 2024-05-24

**Authors:** Toshimichi Onuma, Meshach Asare-Werehene, Yuko Fujita, Yoshio Yoshida, Benjamin K. Tsang

**Affiliations:** 1Department of Obstetrics and Gynecology, Faculty of Medicine, University of Ottawa, Ottawa, ON K1H 8L1, Canada; toonuma@u-fukui.ac.jp (T.O.); masare@ohri.ca (M.A.-W.); 2Interdisciplinary School of Health Sciences, Faculty of Health Sciences, University of Ottawa, Ottawa, ON K1H 8L1, Canada; 3Department of Cellular and Molecular Medicine & The Centre for Infection, Immunity and Inflammation (CI3), Faculty of Medicine, University of Ottawa, Ottawa, ON K1H 8M5, Canada; 4Chronic Disease Program, Ottawa Hospital Research Institute, Ottawa, ON K1H 8L6, Canada; 5Department of Obstetrics and Gynecology, University of Fukui, Fukui 910-1193, Japan; yuchifu@u-fukui.ac.jp; 6Department of Laboratory Medicine and Pathobiology, University of Toronto, Toronto, ON M5S 1A1, Canada

**Keywords:** gynecological cancers, ovarian cancer, plasma gelsolin, chemoresistance, natural killer cells, chemotherapy, tumor microenvironment

## Abstract

Plasma gelsolin (pGSN) overexpression in ovarian cancer (OVCA) disarms immune function, contributing to chemoresistance. The aim of this study was to investigate the immunoregulatory effects of pGSN expression on natural killer (NK) cell function in OVCA. OVCA tissues from primary surgeries underwent immunofluorescent staining of pGSN and the activated NK cell marker natural cytotoxicity triggering receptor 1 to analyze the prognostic impact of pGSN expression and activated NK cell infiltration. The immunoregulatory effects of pGSN on NK cells were assessed using apoptosis assay, cytokine secretion, immune checkpoint-receptor expression, and phosphorylation of STAT3. In OVCA tissue analyses, activated NK cell infiltration provided survival advantages to patients. However, high pGSN expression attenuated the survival benefits of activated NK cell infiltration. In the in vitro experiment, pGSN in OVCA cells induced NK cell death through cell-to-cell contact. pGSN increased T-cell immunoglobulin and mucin-domain-containing-3 expression (TIM-3) on activated NK cells. Further, it decreased interferon-γ production in activated TIM-3+ NK cells, attenuating their anti-tumor effects. Thus, increased pGSN expression suppresses the anti-tumor functions of NK cells. The study provides insights into why immunotherapy is rarely effective in patients with OVCA and suggests novel treatment strategies.

## 1. Introduction

Ovarian cancer (OVCA) has the highest mortality rate among gynecological cancers, primarily due to chemoresistance [1]. Therefore, developing effective treatments for OVCA is vital. Primary debulking surgery is generally performed in patients with advanced-stage OVCA along with platinum-based chemotherapy [1]. However, 80% of OVCA patients develop resistance to platinum-based chemotherapy despite being previously sensitive [2]. Chemoresistance can be acquired by the involvement of DNA repair mechanisms and detoxification proteins (cytochrome P450 complexes) [3,4], cancer stem cells [5], autophagy [6], hypoxia, and endoplasmic reticulum stress [7]. Additionally, cancer cells respond differently to chemotherapy depending on the tumor microenvironment (TME). The immune system is particularly vital for chemotherapy effectiveness [8,9], as tumor infiltration by immune cells has been linked to improved patient responses to chemotherapy and prognosis. OVCA often exhibits poor immune cell infiltration, which makes it a cold tumor, and as a result, immune checkpoint inhibitors are less effective [10,11]. Hence, there is a pressing need to develop novel targets to enhance the response to immunotherapy in patients with OVCA.

The main functions of gelsolin (GSN) include actin filament cleavage, capping, and nucleation, all of which play a crucial role in determining the shape, chemotaxis, and secretion of cells [12,13]. GSN is encoded on human chromosome 9 and has a molecular weight of 80–85 kDa [14,15]. GSN has three main isoforms, namely cytoplasmic GSN (cGSN), secreted/plasma GSN (pGSN), and isoform 3. Among these three well-characterized isoforms, pGSN has the most significant oncogenic and immunosuppressive functions [8,16,17,18,19]. Ovarian and NK cell pGSN mRNA expression levels are comparable according to BioGPS dataset analysis (http://biogps.org, accessed on 28 April 2024) [20] (Appendix A). However, a high level of the pGSN can also be found in malignant tumor cells [19,21]. A previous study has shown that serous ovarian carcinomas exhibit remarkably strong pGSN expression compared with normal fallopian tubes [8]. 

By reducing the cisplatin-induced divergence of the GSN–FLICE-like inhibitory protein (FLIP)–Itch complex, the presence of high pGSN in chemoresistant cells prevents FLIP from being ubiquitinated and degraded, deactivating caspases-8 and -3, and inhibiting GSN cleavage by caspase-3, thereby inhibiting apoptosis. [22]. pGSN is secreted via exosomes. Resistance to chemotherapy is enhanced by exosomal pGSN via the α5β1 integrin/FAK/Akt/HIF-1α axis in chemosensitive OVCA cells [19,23]. In chemoresistant OVCA cells, exosomal pGSN enhances the binding of the promoter region of HIF1α and increases the production of pGSN in exosomes. Thus, positive feedback loops of exosomal pGSN production are generated via the α5β1 integrins/FAK/Akt/HIF1α axis [19,23]. Furthermore, pGSN suppresses the anti-tumor functions of T cells, dendritic cells, and macrophages in the TME. 

It has been demonstrated that patients with OVCA expressing pGSN are less likely to respond to chemotherapy and also have shorter progression-free (PFS) and overall survival (OS) times [8,19,24]. Increased pGSN expression in the OVCA microenvironment renders tumor-infiltrating T cells and M1 macrophages less effective at killing tumors, resulting in worse patient outcomes [8,24]. CD8+ T cells and macrophages infiltrating into OVCA with high pGSN expression produce minimal amounts of interferon γ (IFNγ) and inducible nitric oxide (iNOS) synthase, respectively, and activate caspase-3 to cause cell death [8,24]. IFNγ activates the IFNGR1/JAK/STAT1 pathway, decreasing the intracellular levels of glutathione (GSH) in the OVCA cells. Depletion of CD8+ T cells via pGSN is associated with reduced IFNγ secretion [8]. OVCA cells, therefore, produce more GSH and boost chemoresistance [8]. pGSN increases the expression of IL-4 in CD4+ T cells which are preferentially polarized into type 2 helper T cells [8]. It was found that pGSN selectively attracts and suppresses the viability of M1 macrophages, while not affecting M2 macrophages [24]. Based on these findings, M1 macrophages may be selectively recruited into cancer islets when pGSN levels are high, resulting in reduced viability, while M2 macrophage viability is not affected. pGSN also contributes to chemotherapy resistance by lowering the iNOS level in M1 macrophages, thereby increasing GSH content in OVCA cells [24]. Thus, the M1/M2 ratio is significantly decreased, which is associated with poor survival and chemoresistance [24]. Additionally, pGSN suppresses anti-tumor immune responses by modulating antigen presentation by dendritic cells [25]. Whether this immunosuppression extends to NK cells has yet to be studied. 

Human NK cells account for 15% of all circulating lymphocytes [26] and play a pivotal role in tumor immunity. Cell growth, colony formation, invasion, and PI3K/Akt signaling are inhibited by GSN overexpression in NK cells (YTS cells) [27]. Further investigation is required to identify whether increased pGSN suppresses infiltrated NK cells in the OVCA microenvironment. In some OVCAs, loss of human leukocyte antigen (HLA) class-I expression and deletion of recognizable tumor antigens prevent T-cell activation [28,29,30]. In such cases, NK cell activation may complement reduced T-cell function. Since NK cell-based immunotherapy for OVCA is not established, the activation of NK cells might be achieved by suppressing pGSN, offering new therapeutic targets.

In the current study, OVCA tissues were analyzed for the immunoregulatory roles of pGSN. We further examined if and how pGSN induces checkpoint-receptor expression as well as suppresses NK cell activity. We have demonstrated that increased pGSN inhibits NK cell function by suppressing IFNγ production as well as increasing TIM-3 expression in OVCA chemoresistance.

## 2. Materials and Methods

### 2.1. OVCA Tissue Sampling

This study included 147 patients with epithelial OVCA (high-grade serous (HGS), endometrial, clear cell, and mucinous carcinoma) who received primary surgery between 2006 and 2018 at the University of Fukui Hospital (Fukui, Japan) and Fukui Red Cross Hospital (Fukui, Japan). We obtained written informed consent from all patients, and a certified pathologist diagnosed all tissue samples. Patient characteristics and inclusion and exclusion criteria are provided in Appendix A, respectively. An ethics committee at the University of Fukui Hospital approved the study (IRB number: 20200073) and this study was conducted in adherence to the tenets of the Declaration of Helsinki.

### 2.2. Immunofluorescent Staining for OVCA Tissues

OVCA tissue sections were deparaffinized using Clear PLUS (Falma, Osaka, Japan; catalog number: 306-300-1). Sodium citrate buffer (pH 6.0) was used to immerse the tissue sections. The tissue sections were autoclaved for antigen retrieval, and soaked in 3% hydrogen peroxide solution to attenuate endogenous peroxidase activity. The tissues were then blocked using Blocking One Histo (Nakarai, Tokyo, Japan; catalog number: 06349-64). Sections were incubated with anti-pGSN, anti-natural cytotoxicity receptor 1 (NCR1), and anti-cytokeratin (CK) 8/18 primary antibodies and fluorescent-dye-conjugated secondary antibodies. The Vector TrueVIEW Autofluorescence Quenching Kit with DAPI (Vector Laboratories, Burlingame, CA, USA) was used to suppress autofluorescence while staining the nuclei. The antibodies used are described in detail in Appendix A.

### 2.3. Analyses of Activated NK Cell Infiltration and pGSN Expression in OVCA Tissues

The immunofluorescent-stained tissue sections were scanned using an Olympus FV1200 (Olympus, Tokyo, Japan) with a 20× NA:0.75 objective lens. The scanned image size was 512 × 512 Pixel (634.662 μm × 634.662 μm). The scanned image files were analyzed using QuPath software (version 0.3.0) [31]. Epithelial and stromal areas were determined based on anti-CK8/18 expression; an entire image was automatically divided into epithelial and stromal areas by pixel classification (classifier: artificial neural network (ANN-MLP); resolution: high, 4.97 μm/pixel). The mean fluorescence intensities (MFI) of pGSN and NCR1 were measured for each area.

### 2.4. Messenger RNA Expression Analysis Using The Cancer Genome Atlas (TCGA) Dataset

Messenger RNA analyses were conducted using the Firehouse legacy public dataset available on cBioPortal (https://www.cbioportal.org/; accessed on 1 September 2022). Correlation between pGSN and other markers, namely T-cell immunoreceptor with Ig and ITIM domains (TIGIT), programmed cell death protein 1 (PD-1), lymphocyte-activation gene 3 (LAG-3), cytotoxic T lymphocyte-associated protein 4 (CTLA-4), T-cell immunoglobulin and mucin domain 3 (TIM-3), T-box expressed in T cells (T-BET), and signal transducer and activator of transcription 3 (STAT3), which affected NK cell function [32,33,34], were analyzed using Spearman’s and Pearson’s correlation tests.

### 2.5. Reagents

Etoposide was purchased from TCI America (Portland, OR, USA). pGSN siRNA1 and 2 and scrambled siRNA (control) were purchased from Integrated DNA Technology (Coralville, IA, USA) and Ambion (Foster City, CA, USA). Recombinant human plasma gelsolin (rhpGSN) was designed and generously provided by Dr. Chia-Ching Chang. pGSN cDNA and pCMV3 negative control vectors were purchased from SinoBiological (Beijing, China). UltraComp eBeads Plus compensation beads (Invitrogen, Waltham, MA, USA) were used for compensation. STAT3 Inhibitor C188-9 and Stattic were purchased from Sigma-Aldrich (St. Louis, MO, USA). The list of antibodies used in the cell line experiment is presented in Appendix A.

### 2.6. Cell Lines

NK92MI cells were purchased from ATCC (Manassas, VA, USA) and maintained in MyeloCult™ H5100 (StemCell Technologies; Vancouver, BC, Canada; catalog number: #05150) with penicillin–streptomycin (50 U/mL; Gibco, Waltham, MA, USA; catalog number: 15140-122). HGS cell lines [35,36] (chemosensitive: TOV3041G; chemoresistant: OV90) were generously provided by Anne-Marie Mes-Masson (Centre de Recherche du Centre Hospitalier de l’Université de Montréal (CRCHUM), Montreal, QC, Canada). The HGS cells were maintained in OSE medium (Wisent Inc., Millipore Sigma; St. Louis, MO, USA) with 10% fetal bovine serum (FBS) (Gibco, Waltham, MA, USA; catalog number: 12483-020), amphotericin-B (Wisent Inc., St-Bruno, QC, Canada; catalog number: 450-105-QL), and gentamicin (Wisent Inc., St-Bruno, QC, Canada; catalog number: 450-135-XL). Endometrioid carcinoma cell lines (chemosensitive: A2780S; chemoresistant; A2780CP) were generously provided by Dr. Barbara Vanderhyden. These cells were maintained in Gibco RPMI 1640 (Life Technologies, Grand Island, NY, USA; catalog number: 31800-022) with 10% FBS (Gibco, Waltham, MA, USA; catalog number: 12483-020), penicillin–streptomycin (Gibco, Waltham, MA, USA; catalog number: 15140-122), and amphotericin-B (Gibco, Waltham, MA, USA; catalog number: 15290-026). The details of the histological subtypes and genetic alterations in the cell lines are provided in Appendix A. Chemoresistant (OV90 and A2780CP) cells have more pGSN protein expression than chemosensitive (TOV3041G and A2780S) cells [8,19].

### 2.7. Protein Extraction and WB Analysis

Western blotting (WB) was performed according to previously published methods [19,37]. After protein transfer, the primary antibody in 5% (*w*/*v*) blotto was applied to the membrane, followed by treatment with the appropriate horseradish peroxidase-labeled secondary antibody (1:2000) in 5% (*w*/*v*) blotto. Appendix A provides details about the antibodies. Peroxidase activity was visualized using Pierce ECL Western Blotting Substrate (Thermo Scientific, Rockford, IL, USA) or Amersham ECL Prime Western Blotting Detection Reagent (Cytiva, Vancouver, BC, Canada) with the ChemiDoc imaging system (Bio-Rad, Hercules, CA, USA). The signal intensity generated on the images was measured by densitometry using ImageJ version 1.52 (National Institutes of Health, Bethesda, MD, USA).

### 2.8. Transient Transfection

Chemosensitive OVCA cells A2780S (1.0 × 10^5^) and TOV3041G (0.8 × 10^5^) were seeded into 24 wells and incubated for 24 h. They were transfected with the pGSN cDNA (0.8 μg, 48 h) plasmid and pCMV3 negative control vector (0.8 μg, 48 h) using Lipofectamine 2000 (Invitrogen, Waltham, MA, USA; catalog number: 11668019). WB confirmed the successful overexpression as described previously [19] (Appendix A).

### 2.9. Gene Interference

A2780CP cells (1.25 × 10^5^) were reverse transfected with pGSN siRNAs (30 nM, 48 h) and scrambled sequences (controls; 30 nM, 48 h) using Lipofectamine RNAiMAX (Invitrogen, Waltham, MA, USA; catalog number: 13778150). To exclude off-target effects, two different siRNAs were used to knock down pGSN. This was confirmed by WB as previously reported [19] (Appendix A).

### 2.10. Assessment of Cell Viability

Live cell proliferation was evaluated colorimetrically as an indicator of NK92MI cell viability using the Cell Counting Kit-8 (CCK-8) assay (Sigma-Aldrich, St. Louis, MO, USA) [8]. Staining with trypan blue was performed on NK92MI cells (Gibco, Waltham, MA, USA) to evaluate viability. Cell suspensions were treated with Trypan blue, and cell counts were determined using the TC20 Automated Cell Counter (Bio-Rad, Hercules, CA, USA).

### 2.11. Apoptosis Assay

In the direct co-culture experiment, NK92MI cells were stained with CellTrace Far Red (CTFR) Cell Proliferation Kit (Invitrogen, Waltham, MA, USA) following the manufacturer’s protocol. NK cell apoptosis was assessed using the Annexin V-FITC Apop Kit (Invitrogen, Waltham, MA, USA) as described by the manufacturer’s protocol. NK cells with CTFR were gated separately from the OVCA cells. Apoptosis analyses were conducted in CTFR-labeled NK92MI. Flow cytometry was performed with BD LSRFortessa and FCS Express 7 (De Novo Software, Pasadena, CA, USA).

### 2.12. Intracellular IFNγ Expression Analysis

NK92MI cells were treated with 25 ng/mL PMA, 1 μg/mL ionomycin and rhpGSN or conditioned media (CM) from A2780S cells overexpressing pGSN (Appendix A). A protein transport inhibitor cocktail (Invitrogen, Waltham, MA, USA) was added after 1 h. Cells were washed with phosphate-buffered saline (PBS) after incubation for 6 h. Further, the cells were fixed and permeabilized using the Inside stain kit (Miltenyi Biotec, Bergisch Gladbach, Germany), stained with anti-IFNγ antibody FITC, and analyzed using BD LSRFortessa and FCS express 7 (De Novo Software, Pasadena, CA, USA).

### 2.13. Flow Cytometry Analysis of NK92MI Cells Cocultured with OVCA Cells

To identify dead cells, Zombie NIR dye was applied to the cells following PBS washing. After washing, the cells were stained with dye-conjugated antibodies for cell surface markers. For intracellular staining, they were then fixed and permeabilized with the Inside stain kit (Miltenyi Biotec, Bergisch Gladbach, Germany). CD45 expression was used to analyze NK cells and OVCA cells separately. For phosphorylated STAT3 (pSTAT3) staining, True-Phos Perm Buffer (Biolegend, San Diego, CA, USA) and fixation buffer were used to fix and permeabilize cells. UltraComp eBeads Plus compensation beads (Invitrogen, Waltham, MA, USA) were used for compensation. The cells were analyzed using BD LSRFortessa and FCS Express 7 (De Novo Software, Pasadena, CA, USA). Details of the antibodies used in the cell line experiment are described in Appendix A.

### 2.14. Statistical Analyses

EZR version 1.42 [38] and PRISM software version 9.0 (GraphPad, San Diego, CA, USA) were used for statistical analyses, which included survival analysis, independent samples *t*-test, and one-way analysis of variance with Tukey’s test to determine differences among multiple experimental groups. Two-tailed *p* values <0.05 were considered statistically significant. The relationship between clinicopathological factors was determined using *t*-tests and Pearson’s and Spearman’s correlation tests. Survival curves (PFS and OS) were plotted using the Kaplan–Meier estimator, and *p* values were calculated using the log-rank test. X-tile software (version 3.6.1) was used to determine cutoff values for dividing high or low NCR1 and pGSN expression groups [39]. Hazard ratios and 95% confidence intervals for NCR1, pGSN, residual disease, stage, age, and histological type were evaluated using univariate and multivariate Cox proportional hazards models.

## 3. Results

### 3.1. Patient Characteristics

Tissue samples from patients with epithelial OVCA (N = 147) with different histological subtypes (HGS, endometrioid, clear cell, and mucinous carcinoma) were used in this study. The average age of the patients was 57.0 ± 12.2 years, and the International Federation of Gynecology and Obstetrics (FIGO) stages were as follows: I (N = 74), II (N = 16), III (N = 41), and IV (N = 16) (Appendix A); 68.7% (N = 101) of the patients underwent complete/optimal cytoreduction. The median PFS and OS were 37 and 51 months, respectively (Appendix A).

### 3.2. Increased pGSN Expression Suppressed the Positive Prognostic Effects of NK Cell Infiltration in Patients with OVCA

Increased pGSN expression downregulates T-cell function in the TME [8]. Whether this immunoregulatory role of pGSN extends to NK cells remains to be examined. NCR1 is a receptor that recognizes tumor cells in a non-major histocompatibility complex-restricted manner, which activates NK cells and kills tumor cells [32,33]. NCR1 expression is to date the most reliable marker for NK cells [40]. NK cells express high levels of NCR1 mRNA compared with normal ovaries in BioGPS dataset analysis (http://biogps.org) [20] (Appendix A). Our previous publications, as well as reports from others, have indicated that patient prognosis is dependent on immune cell infiltration and pGSN expression in both epithelial and stromal regions of the tumor [8,24]. Therefore, we assessed the prognostic impact of pGSN expression and NCR1-positive NK cell (NCR1+ NK cell) infiltration into the stroma and epithelial compartments of OVCA tissues (Figure 1A). In general, residual disease is the most important prognostic factor after OVCA surgery [41,42]. We, therefore, investigated the prognostic impact of activated NK cell infiltration on the basis of surgical outcomes. Therefore, we divided the tissue analysis into epithelial and stromal, as well as into residual and non-residual cases.

The cutoff values separating the high and low NCR1 or pGSN expression groups are provided in Appendix A. The relationship between pGSN expression and prognosis is provided in Appendix A. A higher number of NCR1+ NK cells in the epithelial area was associated with better PFS in patients with OVCA compared to the lower number of NCR1+ NK cells in the epithelial area, regardless of their surgical outcome (all cases, *p* = 0.009; non-residual cases, *p* = 0.005; residual cases, *p* = 0.029) (Figure 1B). Similarly, higher numbers of NCR1+ NK cells in the stromal area were associated with improved PFS in the non-residual and residual cases compared to lower number of NCR1+ NK cells in the stromal area (non-residual cases, *p* = 0.003; residual cases, *p* = 0.032) (Figure 1B). These suggest that the infiltration of activated NK cells into the epithelial and stromal areas, regardless of the patient’s surgical outcome, prolongs tumor recurrence. Some of the activated NK cells were negative for pGSN (Appendix A). Interestingly, some of the activated NK cells were positive for pGSN expression, which could be due to uptake, since NK cells have low levels of pGSN [20] (Appendix A and Figure 1C). Patients were then stratified based on pGSN expression and activated NK cell (NCR1+ cells) infiltration into the OVCA tissues. To examine the influence of pGSN on NK cell infiltration, we created the survival curves for pGSN expression in the high NK cell infiltration group (Figure 1D). In the epithelial area of residual cases, patients with high NK cell infiltration with low pGSN expression had significantly better prognoses than high NK cell infiltration with high pGSN expression (*p* = 0.025). However, there was no prognostic impact of pGSN expression in low NK cell infiltration (Appendix A). 

These results suggest that higher pGSN expression in NK cells could trigger an immune suppression mechanism to attenuate their survival benefits. Whether pGSN serves as a chemoattractant for the activation of NK cells in addition to its immunoregulatory role remains to be investigated.

We investigated whether activated NK cell infiltration exerts a different prognostic impact depending on OVCA histological types: NCR1 expression was similar across different histological subtypes in epithelial and stromal areas (Appendix A). In serous and endometrioid cancers, high NCR1 expression in the epithelium was associated with better PFS (serous *p* = 0.018, endometrioid *p* = 0.004, respectively) (Figure 1E). Furthermore, high NCR1 expression in the epithelial area of endometrioid cancer was associated with favorable OS (*p* = 0.0376) (Appendix A). These findings suggest that, in serous and endometrioid cancers, NK cell infiltration had a prognostic impact.

### 3.3. Activated NK Cell Infiltration Is an Independent Prognostic Factor in Residual Cases

A Cox regression analysis was performed to determine independent prognostic factors. In the univariate Cox regression analysis, age ≤56 vs. >56 years, residual disease (RD) ≤1 vs. >1 cm, FIGO stage ≤2 vs. >2, FIGO stage ≤3 vs. >3, and serous vs. non-serous subtypes were associated with PFS in all cases (Appendix A). However, these factors were not associated with PFS when stratified by surgical outcome (Appendix A). High NCR1 expression was associated with better PFS in the epithelial areas in all cases, stromal areas in the non-residual cases, and both epithelial and stromal areas in the residual cases (Appendix A). Additionally, low pGSN levels were associated with prolonged PFS in non-residual cases, but not in all and residual cases (Appendix A). NCR1 and pGSN expression were not associated with OS in the univariate Cox regression analysis (Appendix A). 

In patients with residual disease, tumor biology variables and patient characteristics such as performance status, histological grade, or age have no prognostic impact [41]. Multivariate Cox regression analysis was performed based on the patients’ surgical outcomes. Adjusting for age ≤56 vs. >56 years, FIGO stage ≤3 vs. >3, and histological subtypes, high NCR1 expression was associated with better PFS in the epithelial and stromal areas in residual cases (Table 1). These findings suggest that in residual tumors, activated NK cell infiltration is an independent prognostic factor for predicting prolonged PFS.

### 3.4. Increased pGSN Expression Suppresses IFNγ Production and Induces NK Cell Apoptosis

Tissue pGSN suppresses the survival benefits of activated NK cell infiltration into the TME. To investigate the mechanism involved, NK cells were treated with conditioned media (CM) from chemosensitive (TOV3041G and A2780S) and chemoresistant (OV90 and A2780CP) cells. Regardless of histological subtype differences, CM from chemoresistant OVCA cells significantly suppressed NK cell proliferation compared with their chemosensitive counterparts (Figure 2A). pGSN levels in the CM from A2780S cells with pGSN overexpression were increased compared to A2780S without pGSN overexpression (Appendix A). CM from chemosensitive cells overexpressing pGSN and rhpGSN significantly suppressed NK cell proliferation, and live cell number without a significant increase in apoptosis or caspase-3 activation (Figure 2B–D). Interestingly, CM from chemosensitive cells overexpressing pGSN and rhpGSN significantly decreased IFNγ production in NK cells (Figure 2E). 

Upon direct co-culture, we observed a significant induction of NK cell apoptosis by chemoresistant cells compared with chemosensitive cells depending on the effector (OVCA): target (NK cell) ratio (Figure 3). A high effector ratio induced NK cell apoptosis significantly, which was observed in both HGS (OV90 and TOV3041G) and endometrioid (A2780S and A2780CP) cell lines (Figure 3A). To investigate whether these differences were pGSN-specific, we performed pGSN loss-and-gain-of-function experiments in OVCA cell lines. pGSN overexpression in the chemosensitive cells (TOV3041G and A2780S) induced NK cell apoptosis significantly, while pGSN knockdown in chemoresistant cells (A2780CP) suppressed NK cell apoptosis significantly (Figure 3B,C), suggesting that the immunosuppressive role of pGSN is dependent on OVCA cell-to-NK cell interaction. Given that chemoresistant cells secrete more pGSN compared with their sensitive counterparts [19], we hypothesized that the NK cell death is a result of the increased pGSN content in the NK cells originating from the chemoresistant cells. Previous studies have shown that the uptake of pGSN by immune cells results in immune dysfunction [24]. We have shown that one means of increasing pGSN content in immune cells is through pGSN uptake during co-culture, which could be identified via Western blot, immunofluorescence, or flow cytometry [8,24,25]. We investigated whether pGSN from chemoresistant cells was transmitted to NK cells and suppressed the function of NK cells. NK cells were co-cultured with chemoresistant or chemosensitive cells. Chemoresistant cells have more intracellular pGSN content than chemosensitive cells (Appendix A). We demonstrated that compared with chemosensitive cells, chemoresistant cells significantly increased the content of pGSN in activated NK cells resulting in decreased intracellular IFNγ expression (Figure 3D). This suggests that pGSN derived from chemoresistant cells was transferred to NK cells and suppressed NK cell function.

### 3.5. pGSN Increased TIM-3 Expression in Activated NK Cells

The expression of immune checkpoint receptors causes NK cell dysfunction and increases apoptosis in T cells [43,44]. However, we have yet to investigate if and how pGSN regulates immune checkpoint-receptor expression in NK cells, a mechanism that might contribute to their dysfunction. We, therefore, investigated the OVCA TCGA public dataset to examine if there is a significant association between pGSN mRNA expression and NK cell-related immune checkpoint-receptor genes. Our analysis revealed a significant positive correlation between pGSN mRNA expression and several immune checkpoint receptors (TIGIT, PD-1, LAG-3, CTLA-4, and TIM-3) (Figure 4A). To mechanistically investigate if pGSN regulates the expression of these checkpoint receptors, OVCA cells with pGSN overexpression or knockdown were co-cultured with NK cells, and checkpoint receptors were assessed using flow cytometry (Figure 4B–D). pGSN overexpression in chemosensitive cells significantly increased TIM-3 content in activated NK cells (Figure 4B–C); however, no significant increase in immune checkpoint expression was observed with exogenous rhpGSN and CM from A2780S cells overexpressing pGSN (Appendix A). Although LAG-3 was increased with the endometrioid cell line (A2780S), no significant changes were observed with the HGS cell line (TOV3041G). Moreover, pGSN knockdown in chemoresistant cells significantly decreased only TIM-3 expression in activated NK cells (Figure 4D), suggesting that pGSN-induced expression of TIM-3 is mediated via cell-to-cell contact.

### 3.6. pGSN Suppresses Activated NK Cell Function Relating to pSTAT3 Phosphorylation

To further investigate the immuno-regulatory role of pGSN, OVCA cells with pGSN overexpression or knockdown were directly co-cultured with NK cells, and IFNγ production was analyzed using intracellular flow cytometry. Intracellular pGSN content for chemosensitive cells with pGSN overexpression and chemoresistant cells with pGSN knockdown are provided in Appendix A. pGSN overexpression in chemosensitive cells significantly decreased intracellular IFNγ production in activated TIM-3+ NK cell (Figure 5A), while pGSN knockdown in chemoresistant cells resulted in significantly increased intracellular IFNγ production (Figure 5B). Our TCGA dataset investigation revealed a positive correlation between pGSN expression and galectin-9 and carcinoembryonic antigen-related cell adhesion molecules (CEACAM) -1, which acts as ligands for TIM-3 [45] (Appendix A). In chemoresistant conditions, increased pGSN by OVCA cells increased pGSN and TIM-3 content, resulting in decreased IFNγ production and increased apoptosis in activated NK cells (Figure 5C). On the other hand, pGSN levels were relatively lower in chemosensitive conditions and hence unable to suppress activated NK cell function (Figure 5C). To investigate how pGSN regulates TIM-3 expression and IFNγ production, we investigated the TCGA public dataset to determine potential transcription factors for IFNγ and TIM-3. Our investigation revealed a positive correlation between pGSN mRNA and two transcription factors, signal transducer and activator of transcription 3 (STAT3) and T-box expressed in T cells (T-BET) (Figure 5D and Appendix A). Given that STAT-3 phosphorylation plays a key role in NK cell activity and function [46], STAT-3 phosphorylation was inhibited in the NK cells and the role of pGSN was investigated since NK cell-mediated tumor surveillance is enhanced in the STAT3 knockout mice model [47,48]. pGSN overexpression in the chemosensitive cells resulted in increased pGSN content in TIM3+ activated NK cells (Appendix A). We observed that inhibiting STAT-3 function with C188-9 and Stattic significantly increased IFNγ production in activated TIM-3+ NK cells (Figure 5E). Chemosensitive cells with pGSN overexpression were co-cultured with NK cells and pSTAT-3 levels were measured using flow cytometry. We observed that increasing pGSN significantly inhibited the phosphorylation of STAT-3 in the activated TIM-3+ NK cell population; however, no significant changes were observed in T-BET content (Figure 5F).

## 4. Discussion

This study demonstrated for the first time that pGSN negatively regulates NK cell function in the OVCA microenvironment, causing poor prognosis for patients. Specifically, pGSN expression by OVCA cells results in decreased phosphorylation of STAT-3, reduced IFNγ production, increased expression of TIM-3, and increased apoptosis in activated NK cells. 

OVCA patients with increased infiltration of activated NK cells had improved survival; however, this survival advantage is suppressed when pGSN expression is increased in the cancer cells. The findings of our study are consistent with those of another study that demonstrated that NCR1 knockout mice failed to suppress subcutaneous lymphoma and melanoma [32,49]. This underscores the critical role of activated NK cells in tumor elimination. On the other hand, a previous study observed no improved OS with increased NCR1 expression (activated NK cells) in HGS OVCA [50]. This study found that endometrioid cancer with higher NCR1 expression in the epithelial area had a longer OS. This suggests that the histology subtype may contribute to the prognostic impact of NK cell infiltration. The impact on PFS and OS regarding high NCR1 expression in the stroma area is discordant in all cases or serous cancer cases. High NCR1 expression in the stromal area was associated with worse OS in patients with all cases or serous cancer cases (Appendix A). The high NCR1 expression group had higher pGSN expression in the stromal area of all cases or serous cancer cells compared with the low NCR1 expression group (Appendix A). The group with low NCR1 expression simultaneously had lower pGSN at the OS cutoff point. This suggests that other immune cells, e.g., CD8+ T cells, might compensate for the decrease in NK cells as demonstrated in previous studies [8]. No association between NCR1 expression and progression-free interval (PFI) was observed in all cases or serous cancer cases (Appendix A). Generally, patients with residual disease after primary debulking surgery have poorer outcomes [51]. Gynecologic oncologists are actively seeking the most effective treatment for residual tumors. We believe that the immune system may play an important role in residual cases, which are exclusively treated with chemotherapy. Analysis of the residual cases is important to investigate how chemotherapy and immune infiltration affect patient prognosis. 

Our study found that age, disease stage, and histological subtype were not associated with PFS in patients with residual disease. This was plausible given that the effect of biological factors such as age, performance status, FIGO stage, tumor grade, and presence of ascites on PFS and OS was weaker in residual disease compared to those with complete resection [41]. However, the presence of activated NK cells in the stromal and epithelial areas was independently associated with a better prognosis. FIGO stage (IIIc–IV vs. IIB–IIIB) remained significantly different on multivariate analysis in residual disease [41]. This may be attributed to different patient characteristics and histologic subtypes compared to those in our study. Most OVCA patients are diagnosed at advanced stages [1]. Rates of complete resection in primary debulking surgery were 47.6% for advanced OVCA patients [52]. Managing patients with residual disease in OVCA is challenging. While these typically receive chemotherapy, some do not achieve the desired effect. Therefore, our findings could provide clinicians with valuable information about patients who may potentially benefit from NK cell-based immunotherapy. 

Chemoresistant cells express higher levels of pGSN compared with their chemosensitive counterparts [19]. Upon co-culture with NK cells, we observed an increased NK cell dysfunction, which was evident by decreased IFNγ production, increased TIM-3 expression, and increased apoptosis. These responses were associated with increased pGSN content of OVCA cells. Among all the checkpoint receptors analyzed, only TIM-3 was differentially regulated by pGSN, indicating the selectivity of pGSN in downregulating NK cells. The effect of cancer cells on immune cells sometimes requires cell-to-cell interaction [53,54]. A previous study has shown that reduction of NKG2D does not occur in conditioned media from human hepatoma cell line PLC/PRF/5 but requires direct co-culture with these cells [53]. Although it is unknown how cell-to-cell interaction increases the expression of TIM-3 on NK cells, we hypothesize that a co-stimulatory signal from surface markers could be involved. Our findings are consistent with previous studies that observed that pGSN suppresses IFNγ production as well as induces caspase-3-dependent apoptosis in activated CD8+ T cells and M1 macrophages [8,24]. Tumor infiltration by immune cells is generally associated with improved patient outcomes and suggests that such patients could benefit from immunotherapy. Our findings explain why this might not necessarily be the case for OVCA patients. Although there are activated NK cells in the OVCA microenvironment, we observed that these NK cells were dysfunctional and hence could be misleading in terms of responsiveness to immunotherapy. In addition, this could explain in part why OVCA patients rarely respond to various forms of immunotherapy. To overcome this challenge, tumor levels of pGSN instead of immune cell populations could be used to identify patients with OVCA who may respond to immunotherapy since that could provide a true picture of the tumor microenvironment.

Exogenous recombinant human pGSN and OVCA-derived CM enriched in pGSN could suppress NK cell proliferation. Previous reports have shown that overexpression of pGSN on NK cells inhibits the AKT–PI3K axis and suppresses cell proliferation [27]. Exogenous recombinant human pGSN and OVCA-derived CM enriched in pGSN might be transmitted to NK cells and suppress cell proliferation. On the other hand, previous studies have shown that exogenous recombinant human pGSN and OVCA-derived CM enriched in pGSN could induce caspase-3-dependent death in immune cells [8,24]. This was not the case in our current investigation where pGSN-mediated NK cell dysfunction was only significant when in direct contact with OVCA cells. This suggests that other ligand-receptor interactions could play a role in potentiating the effects of pGSN on NK cells. In the TCGA dataset, pGSN expression correlated positively with galectin-9 and CEACAM-1 (Appendix A). We suspect that the TIM-3 and CEACAM-1/galectin-9 interaction could provide a secondary inhibiting signal potentiating the immunoregulatory effects of pGSN on activated NK cells [45]. TIM-3 and galectin-9 might be deeply involved in NK cell apoptosis by pGSN. The TIM-3 positive CD8+ T cells exhibited more apoptosis than the TIM-3 negative CD8+ T cells [55]. Galectin-9/TIM-3 signaling blockade with anti-TIM-3 antibody reduces apoptosis and, in addition, inhibits tumor growth in mice [55]. Blocking pGSN synthesis as well as the interaction of TIM-3 and gelactin-9 could enhance NK cell activity and function, ultimately reducing apoptosis. This could help effectively kill OVCA cells. T cell and other cell-based therapies have provided only minimal therapeutic effects, hence, NK cells might introduce a new dimension to ovarian cancer treatment offering more potent treatment opportunities [10,11].

While STAT3 is known to reduce the anti-tumor activity of NK cells, it is also reported to be essential for IFNγ production and degranulation of NK cells [56,57]. Early cytokine production, including IFNγ in human NK cells primed with IL-15, requires STAT3 [58]. In STAT-3-deficient mouse NK cells, IFNγ expression was found to be decreased after 4 h of IL-12 stimulation [47]. STAT3 binds directly to the IFNγ promoter and contributes to IFNγ production in mouse NK cells [47]. NK cells treated with STAT3 inhibitor increased IFNγ in response to OVCA cells with pGSN overexpression, suggesting that STAT3 is involved in NK cell dysfunction associated with pGSN. STAT3 may have an altered role depending on the stage of NK cell activation in response to the OVCA cells. Furthermore, other transcription factors may compensate for the STAT3 inhibition, representing a limitation of this study. Flow cytometry-based analysis of the STAT-3 pathway using multiplex panels could provide further clarification. Further studies are warranted.

## 5. Conclusions

Activated NK cell infiltration into OVCA tissues is associated with better outcomes; however, increased pGSN expression in OVCA inhibits this favorable effect. In a chemoresistant condition, increased pGSN by OVCA cells increased pGSN and TIM-3 content, resulting in decreased IFNγ production and increased apoptosis in activated NK cells through cell-to-cell contact. Although these findings are compelling and novel, we aim to validate the immunoregulatory role of pGSN using human primary NK cells as well as animal models in future studies. The role of pGSN-mediated STAT3 pathway in different stages of NK cell activation will be studied. Additionally, the interaction of TIM-3 and CEACAM-1/galectin-9 will be interrupted and the effect of pGSN investigated.

## Figures and Tables

**Figure 1 cells-13-00905-f001:**
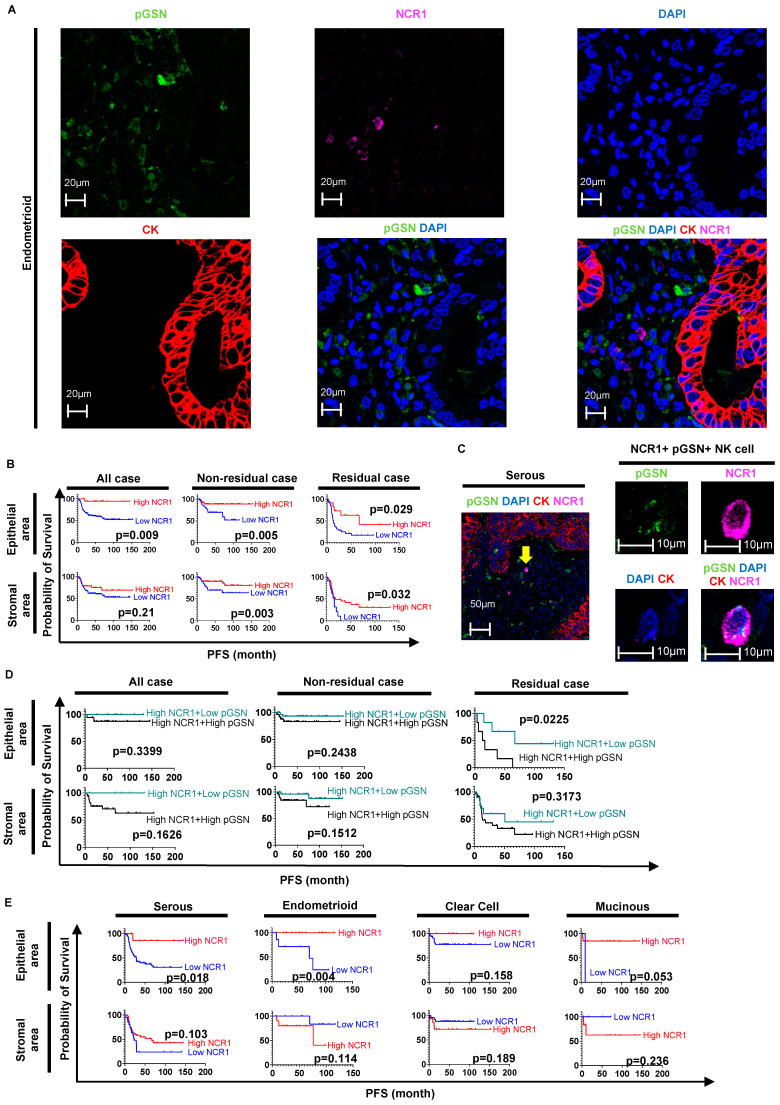
The positive prognostic effect of high NK cell infiltration is suppressed by pGSN overexpression in OVCA tissues. (**A**) OVCA tissues are stained with anti-pGSN (green), anti-NCR1 (pink), anti-cytokeratin 8/18 (red), and DAPI (blue). The MFI is measured. (**B**) Patients are divided into high and low NCR1 expression groups using MFI for PFS analysis. (**C**) Some cells express both pGSN and NCR1 (yellow arrow). (**D**) Patients are divided into high and low pGSN in high NCR1 expression groups. (**E**) Patients are categorized into high and low NCR1 expression groups using MFI for PFS analysis for each OVCA histological subtype. The cut-off values separating the high and low NCR1 or pGSN expression groups are provided in Appendix A. OVCA: ovarian cancer; MFI: mean fluorescent intensity; PFS: progression-free survival; NK: natural killer; NCR1: natural cytotoxicity triggering receptor 1; pGSN: plasma gelsolin.

**Figure 2 cells-13-00905-f002:**
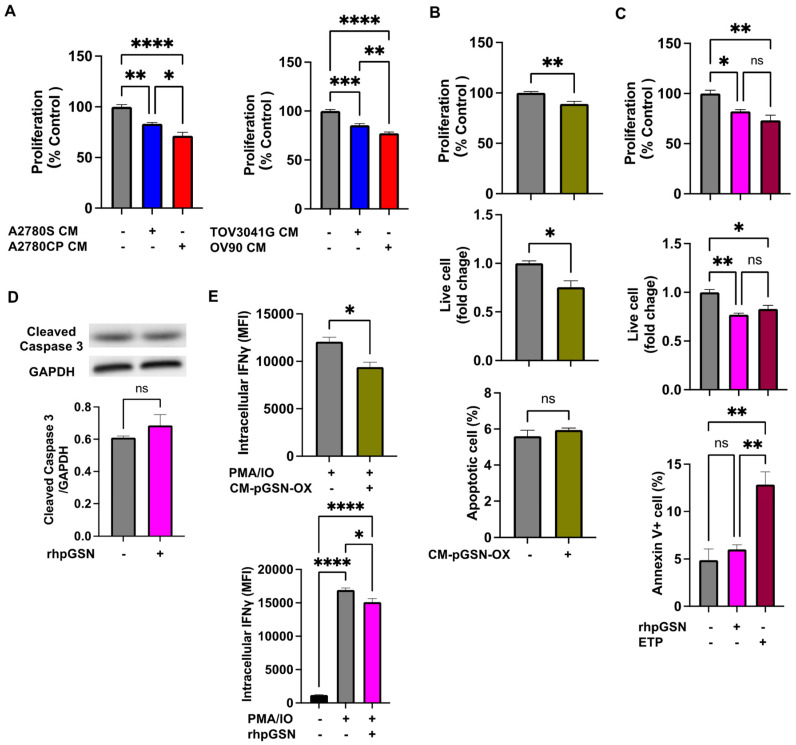
pGSN suppresses cell viability and interferon-γ (IFNγ) production but does not induce NK cell apoptosis without NK cell–OVCA cell contact. (**A**) NK92MI cells (2.0 × 10^4^ cells) were treated with conditioned media (CM) from chemoresistant (A2780CP and OV90) or chemosensitive (A2780S and TOV3041G) cells for 24 h (n = 4). The results were percent cell viability normalized to control. (**B**) NK92MI cells were treated with CM from pGSN-overexpressing A2780S cells (A2780S-CM-pGSN-OX) for 24 h (CCK8 2.0 × 10^4^ cells; n = 4, trypan blue 5.0 × 10^4^ cells; n = 4, apoptosis assay 2.0 × 10^5^ cells; n = 3). The results of the CCK8 assay were percent cell viability normalized to control. The results of the trypan blue assay are shown with the control plasmid as the reference value for fold change. (**C**) NK92MI cells were treated with vehicle, 10 μM rhpGSN, and 6 μM ETP for 12 h (CCK8 2.0 × 10^4^ cells; n = 3, trypan blue 5.0 × 10^4^ cells; n = 3, apoptosis assay 2.0 × 10^5^ cells; n = 3). The results of the CCK8 assay were percent cell viability normalized to control. The results of the trypan blue assay are shown with the vehicle as the reference value for fold change. (**D**) NK92MI cells (5.0 × 10^5^ cells) were treated with 10 μM rhpGSN and vehicle for 12 h for cleaved caspase-3 expression analysis (n = 3). (**E**) NK92MI (5.0 × 10^5^ cells) cells were treated with PMA/ionomycin and 10 μM rhpGSN or A2780S-CM-pGSN-OX for 6 h for intracellular IFNγ analysis (n = 3). These experiment results are expressed as means ± SEM from three or four independent experiments. **** *p* < 0.0001, *** *p* = 0.0001 to 0.001, ** *p* = 0.001 to 0.01, * *p* = 0.01 to 0.05, ns: not significant; NK: natural killer; OVCA: ovarian cancer; ETP: etoposide; pGSN: plasma gelsolin; MFI: median fluorescent intensity.

**Figure 3 cells-13-00905-f003:**
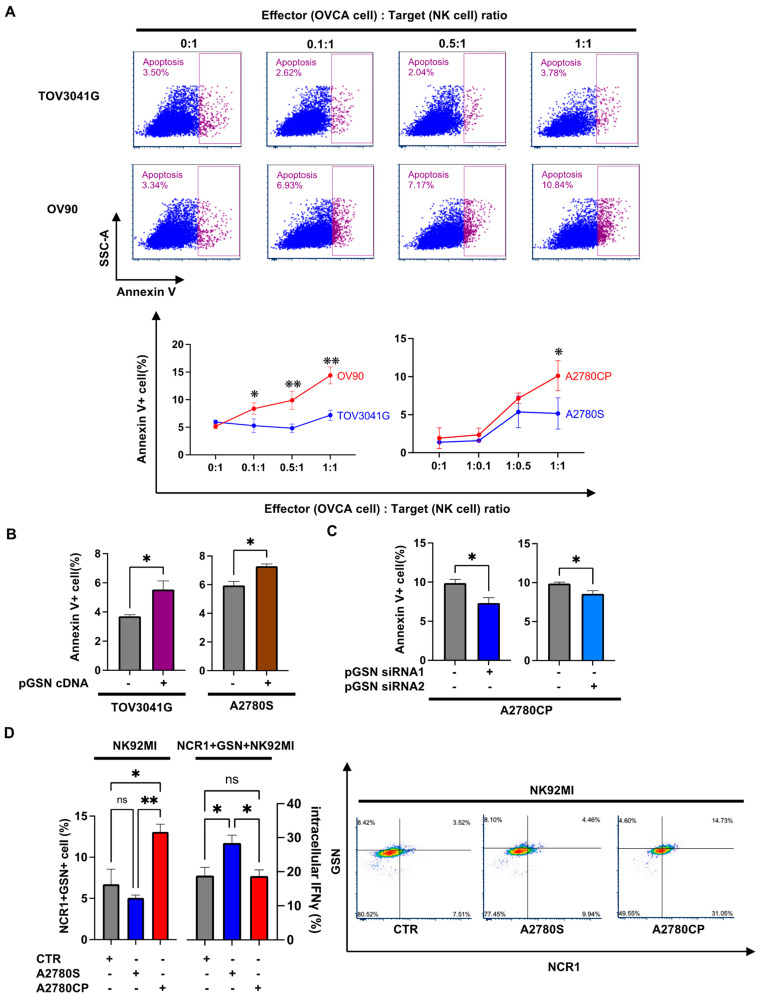
Increased pGSN induces NK cell dysfunction and apoptosis through cell-to-cell contact (**A**) CTFR-labeled NK92MI cells were directly co-cultured with chemoresistant (A2780CP and OV90) or chemosensitive (A2780S and TOV3041G) cells for 5 h and cell death examined using apoptosis assay at different effector: target ratios (n = 3). (**B**) CTFR-labeled NK92MI (2.0 × 10^5^ cells) cells were co-cultured with A2780S cell with pGSN overexpression (A2780S-pGSN-OX) or TOV3041G cell with pGSN overexpression (TOV3041G–pGSN–OX) for 5 h and cell death analyzed using apoptosis assay (n = 3). (**C**) CTFR-labeled NK92MI (2.0 × 10^5^ cells) cells were co-cultured with A2780CP cell with pGSN knockdown using two siRNA (A2780CP–pGSN siRNA1-KD or A2780CP–pGSN siRNA2-KD) for 5 h for apoptosis assay (n = 3). (**D**) OVCA cells (2.0 × 10^5^ cells) were seeded and incubated for 24 h. NK92MI (3.0 × 10^5^ cells) cells were co-cultured with chemosensitive (A2780S) and chemoresistant (A2780CP) cells for 6 h. Intracellular pGSN content in activated NK cells (NCR1+) was analyzed, as well as intracellular interferon-γ (IFNγ) expression (n = 3). The results are expressed as mean ± SEM from three independent experiments. ** *p* = 0.001 to 0.01, * *p* = 0.01 to 0.05, ns: not significant; CTFR: CellTrace Far Red; NK: natural killer; OVCA: ovarian cancer; NCR1: natural cytotoxicity triggering receptor 1; pGSN: plasma gelsolin.

**Figure 4 cells-13-00905-f004:**
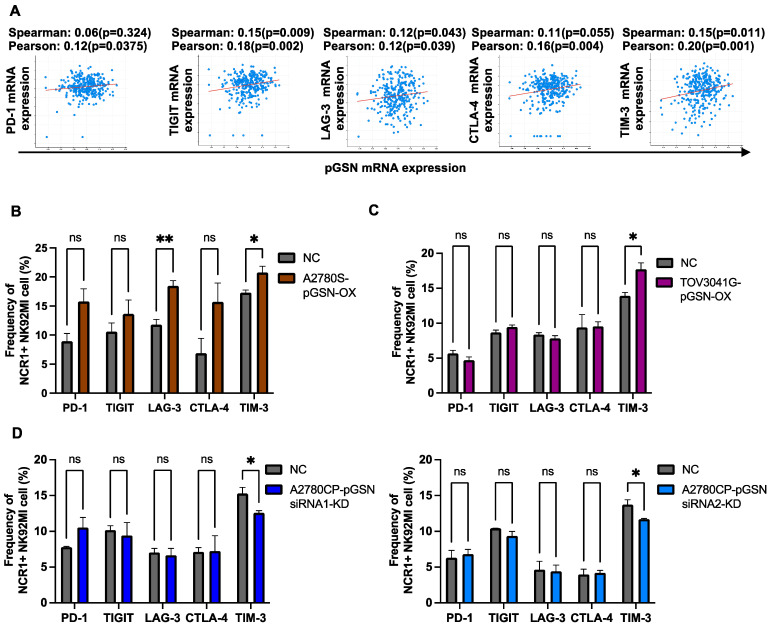
pGSN increases TIM-3 expression in activated NK cells. (**A**) The correlation between pGSN mRNA and mRNA expression of immune checkpoint receptors derived from the TCGA dataset was analyzed using cBioPortal (https://www.cbioportal.org/, accessed on 1 September 2022). (**B**–**D**) NK92MI cells (3.0 × 10^5^ cells) were co-cultured with A2780S cell with pGSN overexpression (A2780S–pGSN–OX), TOV3041G cell with pGSN overexpression (TOV3041G–pGSN–OX), and A2780CP cell with pGSN knockdown using two siRNAs (A2780CP–pGSN siRNA1-KD or A2780CP–pGSN siRNA2-KD) for 6 h. Expression of immune checkpoint receptors (PD-1, TIGIT, LAG-3, CTLA-4, and TIM-3) in activated NK cells (NCR1+) were analyzed using flow cytometry (n = 3). The results are expressed as mean ± SEM from three independent experiments. ** *p* = 0.001 to 0.01, * *p* = 0.01 to 0.05, ns: not significant NK: natural killer; NCR1: natural cytotoxicity triggering receptor 1; pGSN: plasma gelsolin; TIGIT: T-cell immunoreceptor with Ig and ITIM domains; PD-1: programmed cell death protein 1; LAG-3: lymphocyte-activation gene 3; CTLA-4: cytotoxic T lymphocyte-associated protein 4; TIM-3: T-cell immunoglobulin and mucin domain 3.

**Figure 5 cells-13-00905-f005:**
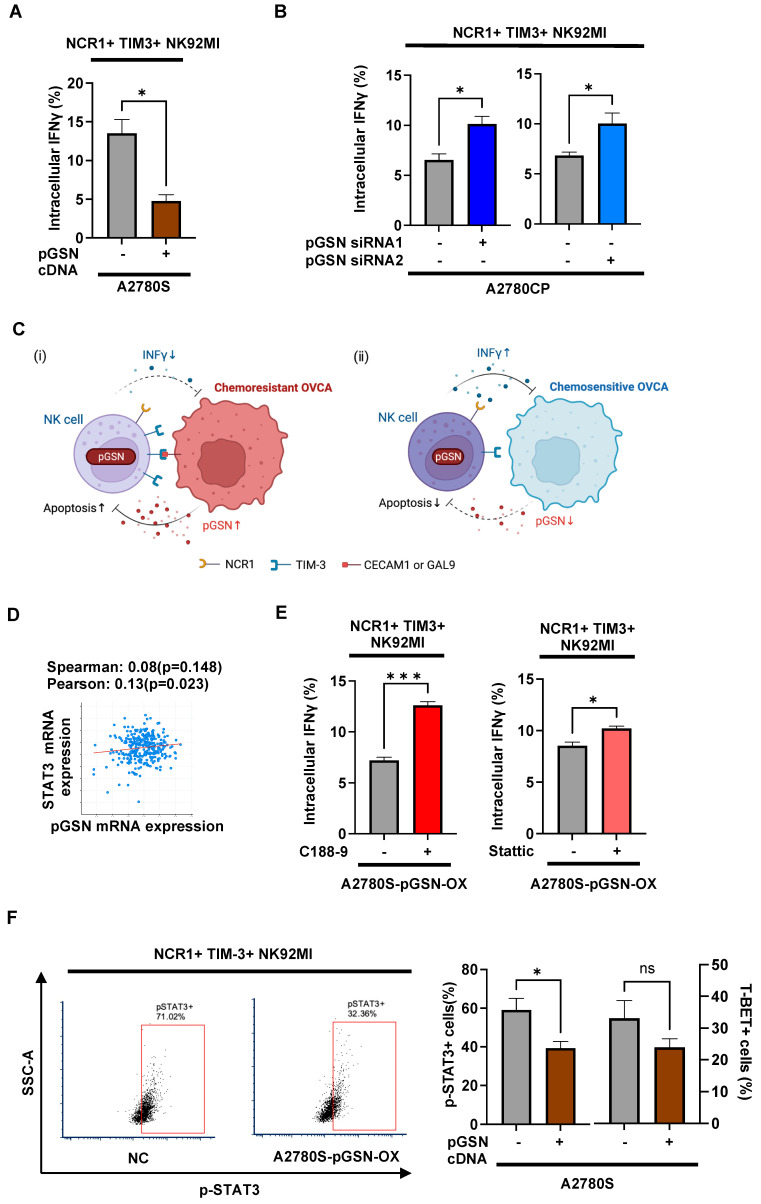
pGSN suppresses activated TIM3+ NK cell function. (**A**,**B**) NK92MI cells (3.0 × 10^5^ cells) were co-cultured with A2780S cell with pGSN overexpression (A2780S–pGSN–OX) or A2780CP cell with pGSN knockdown using two siRNA (A2780CP-siRNA1-KD and A2780CP-siRNA2-KD) for 6 h after which activated NK TIM-3+ cells (NCR1+TIM-3+) positive for interferon-γ (IFNγ) were analyzed using flow cytometry (n = 3). (**C**) In chemoresistant conditions, when NK cells contact with the OVCA, increased pGSN by OVCA cells increased pGSN and TIM-3 content, increased apoptosis, and decreased IFNγ in activated NK cells (i). On the other hand, pGSN levels are relatively lower in chemosensitive conditions and, hence, are unable to suppress activated NK cell function (ii). (**D**) Correlation between pGSN and STAT3 mRNA expressions derived from the TCGA dataset was analyzed using cBioPortal (https://www.cbioportal.org/, accessed on 1 September 2022). (**E**) NK92MI cells were treated with STAT3 inhibitor C188-9 or Sttatic 10μM and vehicle for 1 h. Subsequently, NK92MI cells (3.0 × 10^5^ cells) were co-cultured with A2780S–pGSN–OX for 6 h, after which activated NK TIM-3+ cells (NCR1+TIM-3+) positive for IFNγ were analyzed using flow cytometry (n = 3). (**F**) NK92MI cells (3.0 × 10^5^ cells) were co-cultured with A2780S–pGSN–OX for 6 h, after which activated NK TIM-3+ cells (NCR1+TIM-3+) positive for pSTAT3 and T-BET were analyzed using flow cytometry (n = 3). The results are expressed as mean ± SEM from three independent experiments. *** *p* = 0.0001 to 0.001, * *p* = 0.01 to 0.05, ns: not significant; NK: natural killer; NCR1: natural cytotoxicity triggering receptor 1; pGSN: plasma gelsolin; TIM-3: T-cell immunoglobulin and mucin-domain-containing-3 expression; CEACAM-1: carcinoembryonic antigen-related cell adhesion molecules-1; GAL-9: galactine-9; T-BET: T-box expressed in T cells; pSTAT3: phosphorylated signal transducer and activator of transcription 3.

**Table 1 cells-13-00905-t001:** Multivariable Cox regression analysis for progression-free survival in residual cases.

	NCR1 epi Low vs. High			NCR1 str Low vs. High		
	HR	PFS 95% CI	*p* Value	HR	PFS 95% CI	*p* Value
Model 1	0.3431	0.1166–1.01	0.0521	0.4402	0.1972–0.9824	0.0451
Model 2	0.3115	0.1062–0.9141	0.0337	0.4078	0.1821–0.9131	0.0292

Model 1 was adjusted using age ≤56 vs. >56 years, FIGO stage ≤2 vs. >2, and histology subtypes. Model 2 was adjusted using age ≤56 vs. >56 years, FIGO stage ≤3 vs. >3, and histology subtypes. HR: hazard ratio; epi: epithelial area; str: stromal area; CI: confidence interval; NCR1: natural cytotoxicity triggering receptor 1; PFS: progression-free survival.

## Data Availability

The raw data supporting the conclusions of this article will be made available by the authors without undue reservation.

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
