# Peer review of "Plasma Gelsolin Inhibits Natural Killer Cell Function and Confers Chemoresistance in Epithelial Ovarian Cancer"

_cells, 2024, doi:10.3390/cells13110905_

Round 1
Reviewer 1 Report
Comments and Suggestions for Authors![]()
In this manuscript, Onuma et al. has studied the impact of plasma gelsolin (pGSN) overexpression in the function of natural killer (NK) cells in ovarian cancer (OVCA). The authors have provided both clinical data analysis and performed in vitro experiments to investigate immunoregulatory effects of pGSN. Importantly, the findings from mechanism studies can support the observations from the clinical survival data. Overall, the authors have provided sufficient evidence to fully support the conclusions and the data is consistent with the results.
Specific comments:
1) Figure 1D, what’s the survival curve for patients with Low / High pGSN expression levels regardless of the NCR1 expression status.
2) Figure 5, the statistical significance is missing.
What are the signaling pathways that show different expression levels in Low pGSN and High pGSN groups.
Author Response
Dear Reviewers,
We would like to express our deepest gratitude for your invaluable assistance during the review process of our manuscript. Your thoughtful comments and insightful suggestions have significantly contributed to the improvement of our work.
Your thorough evaluation and constructive feedback have helped us refine the clarity and coherence of our research. We sincerely appreciate the time and expertise you have generously shared with us, as your input has been instrumental in strengthening the overall quality of our manuscript.
We have significantly improved the manuscript to reflect the new changes and suggestions provided as highlighted below:
Reviewer 1:
In this manuscript, Onuma et al. have studied the impact of plasma gelsolin (pGSN) overexpression in the function of natural killer (NK) cells in ovarian cancer (OVCA). The authors have provided both clinical data analysis and performed in vitro experiments to investigate immunoregulatory effects of pGSN. Importantly, the findings from mechanism studies can support the observations from the clinical survival data. Overall, the authors have provided sufficient evidence to fully support the conclusions and the data is consistent with the results.
- Figure 1D, what’s the survival curve for patients with Low / High pGSN expression levels regardless of the NCR1 expression status.
Response: Our previous studies have demonstrated that increased expression of pGSN in OVCA tissues is associated with shortened overall survival and disease-free survival (Asare-Werehene, Nakka, et al. 2020; Onuma et al. 2022). In our current study, we have provided in the supplementary information the survival analysis data for pGSN expression. We have made the following revisions in the manuscript: Line 285, “The relationship between pGSN expression and prognosis is provided in Supplementary Figure S3A”.
- Figure 5, the statistical significance is missing.
Response: Thank you for the correction. Figure 5 has been corrected to show the statistical significance in the clean version of the manuscript.
- What are the signaling pathways that show different expression levels in Low pGSN and High pGSN groups?
Response: Our previous studies and others have demonstrated that increased tissue pGSN is associated with tumor recurrence, chemotherapy, tumor immunosuppression, shortened disease-free survival, and poor overall survival (Asare-Werehene, M, et al., 2020, Oncogene and Cancer Research). Specifically, pGSN induces chemoresistance via autocrine and paracrine pathways by activating the α5β1 integrins/FAK/Akt/HIF‐1α axis in cancer cells especially ovarian cancer cells (Asare-Werehene, et al. 2020, Oncogene and Cancer Research). Additionally, the mechanism of action of pGSN is initiated through small extracellelular vesicles. Silencing pGSN in chemoresistance cells sensitize them to cisplatin induced apoptosis (Asare-Werehene, Nakka, et al. 2020; Onuma et al. 2022). pGSN is involved in the inhibition of CDDP-induced apoptosis in OVCA. FLIP and ITCH form a complex with GSN in the chemosensitive OVCA cell. CDDP dissociates GSN from the GSN‐FLIP‐ITCH complex, leading to FLIP ubiquitination and degradation, caspase-8 and -3 activation, caspase-3 mediated GSN cleavage, and apoptosis (Abedini et al. 2014, PNAS). In chemoresistant OVCA cells, CDDP does not alter the GSN‐FLIP‐ITCH interaction, attenuating its downstream effects (Abedini et al. 2014). In the tumor microenvironment, chemoresistant-derived exosomal pGSN induces CD8+ T cell and M1 macrophage apoptosis via caspase-3 activation as well as polarize naïve CD4 T cells to type 2 helper cells (Asare-Werehene M., et al., 2020, Cancer Research, Cancers).
Line 60: “By reducing cisplatin-induced divergence of the GSN–FLICE-like inhibitory protein (FLIP)–Itch complex, the presence of high pGSN in chemoresistant cells prevents FLIP from being ubiquitinated and degraded, deactivating caspases-8 and -3, and inhibiting GSN cleavage by caspase-3, thereby inhibiting apoptosis. [22]. pGSN is secreted via exosomes. Resistance to chemotherapy is enhanced by exosomal pGSN via the α5β1 integrin/FAK/Akt/HIF-1α axis in chemosensitive OVCA cells [19,23]. In chemoresistant OVCA cells, exosomal pGSN enhances the binding of the promoter region of HIF1α and increases production of pGSN in exosomes. Thus, positive feedback loops of exosomal pGSN production are generated via α5β1 integrins/FAK/Akt/HIF1α axis [19,23]. Fur-thermore, pGSN suppresses the anti-tumor functions of T cells, dendritic cells, and macrophages in the TME. ”
It has been demonstrated that patients with OVCA expressing pGSN are less likely to respond to chemotherapy and also have shorter progression-free (PFS) and overall survival (OS) times [8,19,24]. Increased pGSN expression in the OVCA microenvironment renders tumor-infiltrating T cells and M1 macrophages less effective at killing tumors, resulting in worse patient outcomes [8,24]. CD8+ T cells and macrophages infiltrating into OVCA with high pGSN expression produce minimal amounts of interferon γ (IFNγ) and inducible nitric oxide (iNOS) synthase, respectively, and activates caspase-3 to cause cell death [8,24]. IFNγ activates the IFNGR1/JAK/STAT1 pathway, decreasing the intracel-lular levels of glutathione (GSH) in the OVCA cells. Depletion of CD8+ T cells via pGSN is associated with reduced IFNγ secretion [8]. OVCA cells, therefore, produce more GSH and boost chemoresistance [8]. pGSN increases the expression of IL-4 in CD4+ T cells which are preferentially polarized into type 2 helper T cells [8]. It was found that pGSN selec-tively attracts and suppresses the viability of M1 macrophages, while not affecting M2 macrophages [24]. Based on these findings, M1 macrophages may be selectively recruited into cancer islets when pGSN levels are high, resulting in reduced viability, while M2 macrophage viability is not affected. pGSN also contributes to chemotherapy resistance by lowering the iNOS level in M1 macrophages, thereby increasing GSH content in OVCA cells [24]. Thus, the M1/M2 ratio is significantly decreased, which is associated with poor survival and chemoresistance [24]. Additionally, pGSN suppresses antitumor immune responses by modulating antigen presentation by dendritic cells [25]. Whether this immunosuppression extends to NK cells has yet to be studied.
Reference:
Abedini, Mohammad R., Pei-Wen Wang, Yu-Fang Huang, Mingju Cao, Cheng-Yang Chou, Dar-Bin Shieh, and Benjamin K. Tsang. 2014. “Cell Fate Regulation by Gelsolin in Human Gynecologic Cancers.” Proceedings of the National Academy of Sciences of the United States of America 111 (40): 14442–47. https://doi.org/10.1073/pnas.1401166111.
Asare-Werehene, Meshach, Laudine Communal, Euridice Carmona, Youngjin Han, Yong Sang Song, Dylan Burger, Anne-Marie Mes-Masson, and Benjamin K. Tsang. 2020. “Plasma Gelsolin Inhibits CD8+ T-Cell Function and Regulates Glutathione Production to Confer Chemoresistance in Ovarian Cancer.” Cancer Research 80 (18): 3959–71. https://doi.org/10.1158/0008-5472.CAN-20-0788.
Asare-Werehene, Meshach, Laudine Communal, Euridice Carmona, Tien Le, Diane Provencher, Anne-Marie Mes-Masson, and Benjamin K. Tsang. 2019. “Pre-Operative Circulating Plasma Gelsolin Predicts Residual Disease and Detects Early Stage Ovarian Cancer.” Scientific Reports 9 (1): 13924. https://doi.org/10.1038/s41598-019-50436-1.
Asare-Werehene, Meshach, Kiran Nakka, Arkadiy Reunov, Chen-Tzu Chiu, Wei-Ting Lee, Mohammad R. Abedini, Pei-Wen Wang, et al. 2020. “The Exosome-Mediated Autocrine and Paracrine Actions of Plasma Gelsolin in Ovarian Cancer Chemoresistance.” Oncogene 39 (7): 1600–1616. https://doi.org/10.1038/s41388-019-1087-9.
Asare-Werehene, Meshach, Hideaki Tsuyoshi, Huilin Zhang, Reza Salehi, Chia-Yu Chang, Euridice Carmona, Clifford L. Librach, et al. 2022. “Plasma Gelsolin Confers Chemoresistance in Ovarian Cancer by Resetting the Relative Abundance and Function of Macrophage Subtypes.” Cancers 14 (4). https://doi.org/10.3390/cancers14041039.
Bois, Andreas du, Alexander Reuss, Eric Pujade-Lauraine, Philipp Harter, Isabelle Ray-Coquard, and Jacobus Pfisterer. 2009. “Role of Surgical Outcome as Prognostic Factor in Advanced Epithelial Ovarian Cancer: A Combined Exploratory Analysis of 3 Prospectively Randomized Phase 3 Multicenter Trials: By the Arbeitsgemeinschaft Gynaekologische Onkologie Studiengruppe Ovarialkarzinom (AGO-OVAR) and the Groupe d’Investigateurs Nationaux Pour Les Etudes Des Cancers de l’Ovaire (GINECO).” Cancer 115 (6): 1234–44. https://doi.org/10.1002/cncr.24149.
Chai, Edna Zhi Pei, Muthu K. Shanmugam, Frank Arfuso, Arunasalam Dharmarajan, Chao Wang, Alan Prem Kumar, Ramar Perumal Samy, et al. 2016. “Targeting Transcription Factor STAT3 for Cancer Prevention and Therapy.” Pharmacology & Therapeutics 162 (June): 86–97. https://doi.org/10.1016/j.pharmthera.2015.10.004.
Cooper, M. A., T. A. Fehniger, and M. A. Caligiuri. 2001. “The Biology of Human Natural Killer-Cell Subsets.” Trends in Immunology 22 (11): 633–40. https://doi.org/10.1016/s1471-4906(01)02060-9.
Davies, Richard, Petra Vogelsang, Roland Jonsson, and Silke Appel. 2016. “An Optimized Multiplex Flow Cytometry Protocol for the Analysis of Intracellular Signaling in Peripheral Blood Mononuclear Cells.” Journal of Immunological Methods 436 (September): 58–63. https://doi.org/10.1016/j.jim.2016.06.007.
Fagotti, Anna, Maria Gabriella Ferrandina, Giuseppe Vizzielli, Tina Pasciuto, Francesco Fanfani, Valerio Gallotta, Pasquale Alessandro Margariti, et al. 2020. “Randomized Trial of Primary Debulking Surgery versus Neoadjuvant Chemotherapy for Advanced Epithelial Ovarian Cancer (SCORPION-NCT01461850).” International Journal of Gynecological Cancer: Official Journal of the International Gynecological Cancer Society 30 (11): 1657–64. https://doi.org/10.1136/ijgc-2020-001640.
Guo, Yanwei, Hongqiao Zhang, Xin Xing, Lijuan Wang, Jian Zhang, Lin Yan, Xiaoke Zheng, and Mingzhi Zhang. 2018. “Gelsolin Regulates Proliferation, Apoptosis and Invasion in Natural Killer/T-Cell Lymphoma Cells.” Biology Open 7 (1). https://doi.org/10.1242/bio.027557.
Kang, Chiao-Wen, Avijit Dutta, Li-Yuan Chang, Jayashri Mahalingam, Yung-Chang Lin, Jy-Ming Chiang, Chen-Yu Hsu, et al. 2015. “Apoptosis of Tumor Infiltrating Effector TIM-3+CD8+ T Cells in Colon Cancer.” Scientific Reports 5 (October): 15659. https://doi.org/10.1038/srep15659.
Kuroki, Lindsay, and Saketh R. Guntupalli. 2020. “Treatment of Epithelial Ovarian Cancer.” BMJ 371 (November): m3773. https://doi.org/10.1136/bmj.m3773.
Onuma, Toshimichi, Meshach Asare-Werehene, Yoshio Yoshida, and Benjamin K. Tsang. 2022. “Exosomal Plasma Gelsolin Is an Immunosuppressive Mediator in the Ovarian Tumor Microenvironment and a Determinant of Chemoresistance.” Cells 11 (20). https://doi.org/10.3390/cells11203305.
Wang, Xu, Wenbin Xin, Hua Zhang, Fengmei Zhang, Meilan Gao, Lingling Yuan, Xiaoyan Xu, Xuemei Hu, and Mingdong Zhao. 2014. “Aberrant Expression of P-STAT3 in Peripheral Blood CD4+ and CD8+ T Cells Related to Hepatocellular Carcinoma Development.” Molecular Medicine Reports 10 (5): 2649–56. https://doi.org/10.3892/mmr.2014.2510.
Reviewer 2 Report
Comments and Suggestions for Authors
The study is based on the hypothesis that plasma gelsolin (pGSN) known to be suppressing the anti-tumor functions of T cells, dendritic cells, and macrophages in the tumor microenvironment. The authors wanted to unravel its role in NK cells activity suppression and the mechanistic insight.
The article lacks experimentally validation. The authors must include additional experimentation to validate and enrich the present work.
Points to be considered:
1. Please discuss on the clinical importance of the study.
2. Since there are already reports on the pGSN suppressing the anti-tumor functions of T cells, dendritic cells, and macrophages in the tumor microenvironment, please highlight the novelty of the study towards understanding its similar role in NK cells.
3. Did the authors know about the expression status of both pGSN and NCR1 expression in normal ovarian tissue samples apart from the different types of ovarian cancer shown.
4. Is there any western blot data to show the expression status in different cell types.
5. The authors need to plot the percent cell viability normalized to control for figures 2 A and B. OD values are not recommended to analyze a percent change upon treatment.
6. The western blot for figure 2D doesn’t hold a good resolution.
7. The authors should show the overexpression and the knock down status of the pGSN protein by western blot analysis in the mentioned cell lines, wherever needed.
8. For figure 3A please clarify the experimental details on the co-culture model. The target cells must be stained and gated separately from the effector cells and apoptotic events must be analyzed in the gated target cells population only. There is lack of information in this part in the methods or result section. The flow diagram needs better resolution.
9. The experimental part in figure 3D doesn’t appear clear to understand. Please discuss the experimental set up and the result analysis in more details.
10. The authors should validate the pSTAT3 regulation and its expression using western blot and showing the pSTAT3/STAT3 ratio. Flow analysis only is not the confirmatory observation.
11. The authors should comment on how is the STAT3 pathway regulating the pGSN through proper experimental evidence.
12. The result sections should be discussed in a more detailed way.
13. Please revise on the language and grammatical errors.
Comments on the Quality of English LanguageModerate modification
Author Response
Dear Reviewers,
We would like to express our deepest gratitude for your invaluable assistance during the review process of our manuscript. Your thoughtful comments and insightful suggestions have significantly contributed to the improvement of our work.
Your thorough evaluation and constructive feedback have helped us refine the clarity and coherence of our research. We sincerely appreciate the time and expertise you have generously shared with us, as your input has been instrumental in strengthening the overall quality of our manuscript.
We have significantly improved the manuscript to reflect the new changes and suggestions provided as highlighted below:
Reviewer 2:
The study is based on the hypothesis that plasma gelsolin (pGSN) is known to be suppressing the anti-tumor functions of T cells, dendritic cells, and macrophages in the tumor microenvironment. The authors wanted to unravel its role in NK cells activity suppression and the mechanistic insight.
- Please discuss on the clinical importance of the study.
Response: OVCA has a high mortality rate among other gynecological cancers, mostly due to its resistance to chemotherapy and late diagnosis. Most ovarian cancer patients are diagnosed at advanced stages (Kuroki and Guntupalli 2020). Rates of complete resection were 47.6% for primary debulking surgery (Fagotti et al. 2020). The management of patients with residual disease in OVCA is particularly challenging (du Bois et al. 2009; Asare-Werehene et al. 2019). Chemotherapy is administered to these patients, but some do not achieve the desired effect. Our study may provide clinicians with information regarding patients who may benefit from NK cell-based immunotherapy. The following revisions have been made in the manuscript on line 539: “Most OVCA patients are diagnosed at advanced stages [1]. Rates of complete resection in primary debulking surgery is 47.6% for advanced OVCA patients [52] making the management of patients with residual disease challenging. While these patients typically receive chemotherapy, some do not achieve the desired effect. Additionally, immunotherapy that targets immune cells have achieved modest therapeutic success in OVCA patients unlike other solid tumors. We investigated if and how pGSN suppress the anti-tumor functions of NK cells and how that contributes to chemoresistance. Our findings therefore may provide clinicians with valuable information about patients who may potentially benefit from NK cell-based immunotherapy.
- Since there are already reports on the pGSN suppressing the anti-tumor functions of T cells, dendritic cells, and macrophages in the tumor microenvironment, please highlight the novelty of the study towards understanding its similar role in NK cells.
Response: Thank you for your great suggestion. T cells play a vital role against tumors, which has been investigated in several studies. Human NK cells account for 15% of all circulating lymphocytes and play a pivotal role in tumor immunity (Cooper, Fehniger, and Caligiuri 2001). NK cells can kill targets without pre-sensitization in an MHC-unrestricted manner. NK cell-mediated immunotherapy emerges as a safe and effective treatment approach for any OVCA patients. The success of NK cell therapy depends not only on the ability of immune cells to persist within the complex TME, but also on their ability to maintain their function despite an immunosuppressive environment. This is very essential given current immunotherapies that target T cells, dendritic cells and macrophages have only produced modest therapeutic success in OVCA patients. Since our study revealed that pGSN suppresses NK cell function and induces apoptosis, it is presumed that treatment to suppress pGSN will enhance the efficacy of NK cell-based immunotherapy. This also offers an opportunity to combine NK cell and other immune cell based therapies for maximum therapeutic effects especially in OVCA patients. We have therefore revised the manuscript to reflect the following changes as located on line 579: “We suspect that the TIM-3 and CEACAM-1/galectin-9 interact could provide a secondary inhibiting signal potentiating the immunoregulatory effects of pGSN on activated NK cells [45]. TIM-3 and galectin-9 might be deeply involved in NK cell apoptosis by pGSN. The TIM-3 positive CD8+ T cells exhibited more apoptosis than the TIM-3 negative CD8+ T cells [55]. Galectin-9/TIM-3 signaling blockade with anti-TIM-3 antibody reduces the apoptosis and in addition, inhibits tumor growth in mice [55]. Blocking pGSN synthesis as well as the interaction of TIM-3 and gelactin-9 could enhance NK cell activity and function, ultimately reducing apoptosis. This could help effectively kill OVCA cells. T cell and other cell-based therapies have provided only minimal therapeutic effects hence NK cells might introduce a new dimension to ovarian cancer treatment offering a more potent treatment opportunities [10,11].”
- Did the authors know about the expression status of both pGSN and NCR1 expression in normal ovarian tissue samples apart from the different types of ovarian cancer shown.
Response: We have previously demonstrated that pGSN is minimally expressed in normal fallopian tube tissues compared with OVCA tissues (Asare-Werehene M, et al., 2020, Cancer Research). In BioGPS dataset analysis, NK cells express high levels of NCR1 mRNA compared with normal ovaries. Meanwhile, pGSN mRNA expression is low in normal ovaries and NK cells. The following revisions have been made to following lines in the manuscript: “line 262, “NK cells express high levels of NCR1 mRNA compared with normal ovaries in BioGPS dataset analysis (http://biogps.org) [20] (Supplementary Figure S1B)”. Line 55: “Ovarian and NK cell pGSN mRNA expression levels are comparable according to BioGPS dataset analysis (http://biogps.org) [20] (Supplementary Figure S1A). However, a high level of pGSN can also be found in malignant tumor cells [19,21]. A previous study has shown that serous ovarian carcinomas exhibit remarkably strong pGSN expression compared with normal fallopian tubes [8].”
- Is there any western blot data to show the expression status in different cell types.
Response: Flow cytometry and immunostaining were used to characterize the expression status of pGSN and NCR1 in OVCA cells and tissues. Previously, Western blot was used to characterize our chemoresistant and sensitive OVCA cells as well as tissues (Asare-Werehene, M, et al., 2020, Cancer Research). In general, chemoresistant (OV90 and A2780CP) cells have more pGSN content compared with their chemosensitive counterparts (TOV3041G and A2780S) cells.
- The authors need to plot the percent cell viability normalized to control for figures 2 A and B. OD values are not recommended to analyze a percent change upon treatment.
Response: As recommended by the reviewer, we have revised figure 2 with percent changes normalized to the control.
- The western blot for figure 2D doesn’t hold a good resolution.
Response: As suggested by the reviewer, figure 2D has been updated with better resolution images.
- The authors should show the overexpression and the knock down status of the pGSN protein by western blot analysis in the mentioned cell lines, wherever needed.
Response: As shown in Supplementary Figure S2A-C, both overexpression and knockdown of the pGSN protein in OVCA cella were confirmed by western blot analysis. In direct co-culture experiments, WB is extremely challenging. The signal obtained by WB represents the total amount of extracted protein. Unfortunately, separate signal detection of OVCA cell cannot be accomplished by WB. Rather, we used flow cytometry to assess changes in pGSN after co-culture conditions (Supplementary Figure S5A and S7A,B).
- For figure 3A please clarify the experimental details on the co-culture model. The target cells must be stained and gated separately from the effector cells and apoptotic events must be analyzed in the gated target cells population only. There is lack of information in this part in the methods or result section. The flow diagram needs better resolution.
Response: We agree with the reviewer’s comment and have provided sufficient explanation in the method section as well as the legend of Figure 3A. The NK cells were stained with Cell Trace Far Red (CTFR). NK cells with CTFR were gated separately from the OVCA cells. All apoptotic events were analyzed in gated NK cells. We added the details in the Method section. The flow diagrams have been replaces with higher resolution versions. The following revisions have been made: Line 208, “In direct co-culture experiment, NK92MI cells were stained with Cell Trace Far Red Cell (CTFR) Proliferation Kit (Invitrogen, Waltham, MA, USA) following the manufacturer’s protocol. NK cell apoptosis was assessed using the Annexin V-FITC Apop Kit (Invitrogen, Waltham, MA, USA) as described by the manufacturer’s protocol. NK cells with CTFR were gated separately from the OVCA cells. Apoptosis analyses were conducted in CTFR-labeled NK92MI. Flow cytometry was performed with BD LSRFortessa and FCS express 7 (De Novo Software, Pasadena, CA, USA).”
- The experimental part in figure 3D doesn’t appear clear to understand. Please discuss the experimental set up and the result analysis in more details.
Response: We investigated whether chemoresistant cell-derived pGSN induced NK cell suppression compared with their sensitive counterparts. In Figure 3D, NK cells were co-cultured with chemoresistant or chemosensituve OVCA cells. CD45+ NCR1+ cells which indicated activated NK cells were gated and analyzed. We observed that upon co-culturing with chemoresistant cells, pGSN content in activated NK cells increased resulting in decreased IFNγ compared with chemosensitive cells. The following revisions have been made for clarity: Line 366, “Given that chemoresistant cells secrete more pGSN compared with their sensitive counterparts [19], we hypothesized that the NK-cell death is a result of the increased pGSN content in the NK cells originating from the chemoresistant cells. Previous studies have shown that uptake of pGSN by immune cells result in immune dysfunction [24]. We have shown that one means of increasing pGSN content in immune cells is through pGSN uptake during co-culture, which could be identified via western blot, immunofluores-cence, or flow cytometry [8,24,25]. We investigated whether pGSN from chemoresistant cells was transmitted to NK cells and suppressed the function of NK cells. NK cells were co-cultured with chemoresistant or chemosensitive cells. Chemoresistant cells have more intracellular pGSN content than chemosensitive cells (Supplemetary Figure S5). We demonstrated that compared with chemosensitive cells, chemoresistant cells significantly increased the content of pGSN in activated NK cells resulting in decreased intracellular IFNγ expression (Figure 3D). This suggests that pGSN derived from chemoresistant cells was transferred to NK cells and suppressed NK cell function.”
- The authors should validate the pSTAT3 regulation and its expression using western blot and showing the pSTAT3/STAT3 ratio. Flow analysis only is not the confirmatory observation.
Response: Performing Western blot on a direct co-culture will be challenging since the protein extracted will be derived from both cell populations. Single cells can be analyzed using flow cytometry. CD45+ NCR1+ cells which indicated activated NK cells were gated and analyzed in a coculture setting. pSTAT3 is the activated state of STAT3 (Chai et al. 2016). A previous coculture setting study measured pSTAT3 expression on CD8+ T cells in PBMCs cocultured with hepatocellular carcinoma cell lines (Wang et al. 2014). Phosphorylation of STAT3 is generally a quick reaction. Flowcytometry based pathway analysis has been proposed to analyze only phosphorylated proteins (Davies et al. 2016). For this reason, we analyzed only pSTAT3 on activated NK cells using flowcytometry.
- The authors should comment on how is the STAT3 pathway regulating the pGSN through proper experimental evidence.
Response: Our results demonstrate for the first time, the involvement of STAT3 phosphorylation and pGSN interaction in NK cell function. Detailed mechanistic interaction of STAT3 and pGSN will be considered in future studies including animal models. Regardless, we used multiplex panels for flow cytometry targeting the STAT3 pathway to elaborate on the proposed pathway. This limitation is acknowledged in our study as stated in the manuscript: Line 595, “NK cells treated with STAT3 inhibitor increased IFNγ in response to OVCA cell with pGSN overexpression, suggesting that STAT3 is involved in NK cell dysfunction asso-ciated with pGSN. STAT3 may have an altered role depending on the stage of NK cell activation in response to the OVCA cells. Furthermore, other transcription factors may compensate for the STAT3 inhibition, representing a limitation of this study. Flow cy-tometry-based analysis of the STAT-3 pathway using multiplex panels could provide further clarification. Further studies are warranted.”
- The result sections should be discussed in a more detailed way.
Response: We have added a detailed description of the experimental results so that readers can better appreciate our study. The following revisions have been made:
3.1. Patient characteristics
Tissue samples from patients with epithelial OVCA (N=147) with different histo-logical subtypes (HGS, endometrioid, clear cell, and mucinous carcinoma) were used in this study. The average age of the patients was 57.0±12.2 years, and the International Federation of Gynecology and Obstetrics (FIGO) stages were as follows: I (N=74), II (N=16), III (N=41), and IV (N=16) (Supplementary Table S1); 68.7% (N=101) of the patients underwent complete/optimal cytoreduction. The median PFS and OS were 37 and 51 months, respectively (Supplementary Table S1).
3.2. Increased pGSN expression suppressed the positive prognostic effects of NK cell infiltration in patients with OVCA
Increased pGSN expression downregulates T-cell function in the TME [8]. Whether this immunoregulatory role of pGSN extends to NK cells remains to be examined. NCR1 is a receptor that recognizes tumor cells in a non-major histocompatibility com-plex-restricted manner, which activates NK cells and kills tumor cells [32,33]. NCR1 expression is to date the most reliable marker for NK cells [40]. NK cells express high levels of NCR1 mRNA compared with normal ovaries in BioGPS dataset analysis (http://biogps.org) [20] (Supplementary Figure S1B). Our previous publications, as well as reports from others, have indicated that patient prognosis is dependent on immune cell infiltration and pGSN expression in both epithelial and stromal regions of the tumor [8,24]. Therefore, we assessed the prognostic impact of pGSN expression and NCR1-positive NK cell (NCR1+ NK cell) infiltration into the stroma and epithelial compartments of OVCA tissues (Figure 1A). In general, residual disease is the most important prognostic factor after OVCA surgery [41,42]. We therefore investigated the prognostic impact of activated NK cell infiltration on the basis of surgical outcome. Therefore, we divided the tissue analysis into epithelial and stromal, as well as into residual and non-residual cases.
The cut-off values separating the high and low NCR1 or pGSN expression groups are provided in Supplementary Tables S5–10. The relationship between pGSN expression and prognosis is provided in Supplementary Figure S3A. A higher number of NCR1+ NK cells in the epithelial area was associated with better PFS in patients with OVCA compared to lower number of NCR1+ NK cells in the epithelial area, regardless of their surgical outcome (All cases, p=0.009; Non-residual cases, p=0.005; Residual cases, p=0.029) (Figure 1B). Similarly, higher numbers of NCR1+ NK cells in the stromal area were associated with improved PFS in the non-residual and residual cases compared to lower number of NCR1+ NK cells in stromal area (Non-residual cases, p=0.003; Residual cases, p=0.032) (Figure 1B). These suggest that the infiltration of activated NK cells into the epithelial and stromal areas, regardless of the patient’s surgical outcome, prolongs tumor recurrence. Some of the activated NK cells were negative for pGSN (Supplemetary Figure S3B). In-terestingly, some of the activated NK cells were positive for pGSN expression, which could be due to uptake, since NK cells have low levels of pGSN [20] (Supplementary Figure S1B and Figure 1C). Patients were then stratified based on pGSN expression and activated NK cell (NCR1+ cells) infiltration into the OVCA tissues. To examine the influence of pGSN on NK cell infiltration, we created the survival curves for pGSN expression in high NK cell infiltration group (Figure 1D). In the epithelial area of residual cases, patients with high NK cell infiltration with low pGSN expression had significantly better prognosis than high NK cell infiltration with high pGSN expression (p=0.025). However, there was no prognostic impact of pGSN expression in low NK cell infiltration (Supplementary Figure 3C).
These results suggest that higher pGSN expression in NK cells could trigger an immune suppression mechanism to attenuate their survival benefits. Whether pGSN serves as a chemoattractant for the activation of NK cells in addition to its immunoreg-ulatory role remains to be investigated.
We investigated whether activated NK cell infiltration exert a different prognostic impact depending on OVCA histological types: NCR1 expression was similar across different histological subtypes in epithelial and stromal areas (Supplementary Figure S4A). In serous and endometrioid cancers, high NCR1 expression in the epithelium was associated with better PFS (Serous p=0.018, Endometrioid p=0.004, respectively) (Figure 1E). Furthermore, high NCR1 expression in the epithelial area of endometrioid cancer was associated with favorable OS (p=0.0376) (Supplementary Figure S4B). These findings suggest that, in serous and endometrioid cancers, NK cell infiltration had a prognostic impact.
3.3. Activated NK cell infiltration is an independent prognostic factor in residual cases
A Cox regression analysis was performed to determine independent prognostic factors. In the univariate Cox regression analysis, age ≤56 vs. >56 years, residual disease (RD) ≤1 vs. >1 cm, FIGO stage ≤2 vs. >2, FIGO stage ≤3 vs. >3, and serous vs. non-serous subtype were associated with PFS in all cases (Supplementary Table S11). However, these factors were not associated with PFS when stratified by surgical outcome (Supplementary Table S11). High NCR1 expression was associated with better PFS in the epithelial areas in all cases, stromal areas in the non-residual cases, and both epithelial and stromal areas in the residual cases (Supplementary Table S11). Additionally, low pGSN levels were associated with prolonged PFS in non-residual cases, but not in all and residual cases (Supplementary Table S11). NCR1 and pGSN expression were not associated with OS in the univariate Cox regression analysis (Supplementary Table S12).
In patients with residual disease, tumor biology variables and patient characteristics such as performance status, histological grade, or age have no prognostic impact [41]. Multivariate Cox regression analysis was performed based on the patients’ surgical outcomes. Adjusting for age ≤56 vs. >56 years, FIGO stage ≤3 vs. >3, and histological subtypes, high NCR1 expression was associated with better PFS in the epithelial and stromal areas in residual cases (Table 1). These findings suggest that in residual tumors, activated NK cell infiltration is an independent prognostic factor for predicting prolonged PFS.
3.4. Increased pGSN expression suppresses IFNγ production and induces NK cell apoptosis
Tissue pGSN suppresses the survival benefits of activated NK cell infiltration into the TME. To investigate the mechanism involved, NK cells were treated with conditioned media (CM) from chemosensitive (TOV3041G and A2780S) and chemoresistant (OV90 and A2780CP) cells. Regardless of histological subtype differences, CM from chemoresistant OVCA cells significantly suppressed NK cell proliferation compared with their chemo-sensitive counterparts (Figure 2A). pGSN levels in the CM from A2780S cells with pGSN overexpression was increased compared to A2780S without pGSN overexpression (Sup-plementary Figure S2D). CM from chemosensitive cells overexpressing pGSN and rhpGSN significantly suppressed NK cell proliferation, live cell number without a sig-nificant increase in apoptosis or caspase-3 activation (Figure 2B–D). Interestingly, CM from chemosensitive cells overexpressing pGSN and rhpGSN significantly decreased IFNγ production in NK cells (Figure 2E).
Upon direct co-culture, we observed a significant induction of NK cell apoptosis by chemoresistant cells compared with chemosensitive cells depending on the effector (OVCA): target (NK cell) ratio (Figure 3). High effector ratio induced NK cell apoptosis significantly, which was observed in both HGS (OV90 and TOV3041G) and endometrioid (A2780S and A2780CP) cell lines (Figure 3A). To investigate whether these differences were pGSN specific, we performed pGSN loss-and-gain-of-function experiments in OVCA cell lines. pGSN overexpression in the chemosensitive cells (TOV3041G and A2780S) induced NK-cell apoptosis significantly, while pGSN knockdown in chemoresistant cells (A2780CP) suppressed NK cell apoptosis significantly (Figure 3B and C), suggesting that the immunosuppressive role of pGSN is dependent on OVCA cell-to-NK cell interaction. Given that chemoresistant cells secrete more pGSN compared with their sensitive counterparts [19], we hypothesized that the NK-cell death is a result of the increased pGSN content in the NK cells originating from the chemoresistant cells. Previous studies have shown that uptake of pGSN by immune cells result in immune dysfunction [24]. We have shown that one means of increasing pGSN content in immune cells is through pGSN uptake during co-culture, which could be identified via western blot, immunofluores-cence, or flow cytometry [8,24,25]. We investigated whether pGSN from chemoresistant cells was transmitted to NK cells and suppressed the function of NK cells. NK cells were co-cultured with chemoresistant or chemosensitive cells. Chemoresistant cells have more intracellular pGSN content than chemosensitive cells (Supplemetary Figure S5). We demonstrated that compared with chemosensitive cells, chemoresistant cells significantly increased the content of pGSN in activated NK cells resulting in decreased intracellular IFNγ expression (Figure 3D). This suggests that pGSN derived from chemoresistant cells was transferred to NK cells and suppressed NK cell function.
3.5. pGSN increased TIM-3 expression in activated NK cells
The expression of immune checkpoint receptors causes NK cell dysfunction and increases apoptosis in T-cells [43,44]. However, we are yet to investigate if and how pGSN regulate immune checkpoint receptor expression in NK cells, a mechanism that might contribute to their dysfunction. We, therefore investigated the OVCA TCGA public dataset to examine if there is a significant association between pGSN mRNA expression and NK cell-related immune checkpoint receptor genes. Our analysis revealed a sig-nificant positive correlation between pGSN mRNA expression and several immune checkpoint receptors (TIGIT, PD-1, LAG-3, CTLA-4, and TIM-3) (Figure 4A). To mecha-nistically investigate if pGSN regulates the expression of these checkpoint receptors, OVCA cells with pGSN overexpression or knockdown were co-cultured with NK cells, and checkpoint receptors were assessed using flow cytometry (Figure 4B–D). pGSN overex-pression in chemosensitive cells significantly increased TIM-3 content in activated NK cells (Figure 4B-C); however, no significant increase in immune checkpoint expression was observed with exogenous rhpGSN and CM from A2780S cells overexpressing pGSN (Supplementary Figure S6A and B). Although LAG-3 was increased with endometrioid cell line (A2780S), no significant changes were observed with HGS cell line (TOV3041G). Moreover, pGSN knockdown in chemoresistant cells significantly decreased only TIM-3 expression in activated NK cells (Figure 4D), suggesting that pGSN-induced expression of TIM-3 is mediated via cell-to-cell contact.
3.6. pGSN suppresses activated NK cell function relating to pSTAT3 phosphorylation
To further investigate the immuno-regulatory role of pGSN, OVCA cells with pGSN overexpression or knockdown were directly co-cultured with NK cells, and IFNγ pro-duction was analyzed using intracellular flow cytometry. Intracellular pGSN content for chemosensitive cell with pGSN overexpression and chemoresistant cell with pGSN knockdown were provided in Supplementary Figure S7A and B. pGSN overexpression in chemosensitive cell significantly decreased intracellular IFNγ production in activated TIM-3+ NK cell (Figure 5A), while pGSN knock-down in chemoresistant cell resulted in significantly increased intracellular IFNγ production (Figure 5B). Our TCGA dataset investigation revealed a positive correlation between pGSN expression and galectin-9 and carcinoembryonic antigen-related cell adhesion molecules (CEACAM) -1, which acts as ligands for TIM-3 [45] (Supplementary Figure S8). In chemoresistant condition, increased pGSN by OVCA cells increased pGSN and TIM-3 content, resulting in decreased IFNγ production and increased apoptosis in activated NK cells (Figure 5C). On the other hand, pGSN levels were relatively lower in chemosensitive conditions and hence unable to suppress activated NK cell function (Figure 5C). To investigate how pGSN regulates TIM-3 expression and IFNγ production, we investigated TCGA public dataset to determine potential transcription factors for IFNγ and TIM-3. Our investigation revealed a positive correlation between pGSN mRNA and two transcription factors, signal transducer and activator of transcription 3 (STAT3) and T-box expressed in T cells (T-BET) (Figure 5D and Supplementary Figure S9). Given that STAT-3 phosphorylation plays a key role in NK cell activity and function [46], STAT-3 phosphorylation was inhibited in the NK cells and the role of pGSN was investigated since NK cell-mediated tumor surveillance is enhanced in STAT3 knockout mice model [47,48]. pGSN overexpression in the chemosensitive cells resulted in increased pGSN content in TIM3+ activated NK cells (Supplementary Figure S9). We observed that inhibiting STAT-3 function with C188-9 and Stattic significantly increased IFNγ production in activated TIM-3+ NK cells (Figure 5E). Chemosensitive cells with pGSN overexpression were co-cultured with NK cells and pSTAT-3 levels were measured using flow cytometry. We observed that increasing pGSN significantly in-hibited the phosphorylation of STAT-3 in the activated TIM-3+ NK cell population; however, no significant changes were observed in T-BET content (Figure 5F).
- Please revise on the language and grammatical errors.
Response: We have revised and improved the English language across the manuscript.
Reference:
Abedini, Mohammad R., Pei-Wen Wang, Yu-Fang Huang, Mingju Cao, Cheng-Yang Chou, Dar-Bin Shieh, and Benjamin K. Tsang. 2014. “Cell Fate Regulation by Gelsolin in Human Gynecologic Cancers.” Proceedings of the National Academy of Sciences of the United States of America 111 (40): 14442–47. https://doi.org/10.1073/pnas.1401166111.
Asare-Werehene, Meshach, Laudine Communal, Euridice Carmona, Youngjin Han, Yong Sang Song, Dylan Burger, Anne-Marie Mes-Masson, and Benjamin K. Tsang. 2020. “Plasma Gelsolin Inhibits CD8+ T-Cell Function and Regulates Glutathione Production to Confer Chemoresistance in Ovarian Cancer.” Cancer Research 80 (18): 3959–71. https://doi.org/10.1158/0008-5472.CAN-20-0788.
Asare-Werehene, Meshach, Laudine Communal, Euridice Carmona, Tien Le, Diane Provencher, Anne-Marie Mes-Masson, and Benjamin K. Tsang. 2019. “Pre-Operative Circulating Plasma Gelsolin Predicts Residual Disease and Detects Early Stage Ovarian Cancer.” Scientific Reports 9 (1): 13924. https://doi.org/10.1038/s41598-019-50436-1.
Asare-Werehene, Meshach, Kiran Nakka, Arkadiy Reunov, Chen-Tzu Chiu, Wei-Ting Lee, Mohammad R. Abedini, Pei-Wen Wang, et al. 2020. “The Exosome-Mediated Autocrine and Paracrine Actions of Plasma Gelsolin in Ovarian Cancer Chemoresistance.” Oncogene 39 (7): 1600–1616. https://doi.org/10.1038/s41388-019-1087-9.
Asare-Werehene, Meshach, Hideaki Tsuyoshi, Huilin Zhang, Reza Salehi, Chia-Yu Chang, Euridice Carmona, Clifford L. Librach, et al. 2022. “Plasma Gelsolin Confers Chemoresistance in Ovarian Cancer by Resetting the Relative Abundance and Function of Macrophage Subtypes.” Cancers 14 (4). https://doi.org/10.3390/cancers14041039.
Bois, Andreas du, Alexander Reuss, Eric Pujade-Lauraine, Philipp Harter, Isabelle Ray-Coquard, and Jacobus Pfisterer. 2009. “Role of Surgical Outcome as Prognostic Factor in Advanced Epithelial Ovarian Cancer: A Combined Exploratory Analysis of 3 Prospectively Randomized Phase 3 Multicenter Trials: By the Arbeitsgemeinschaft Gynaekologische Onkologie Studiengruppe Ovarialkarzinom (AGO-OVAR) and the Groupe d’Investigateurs Nationaux Pour Les Etudes Des Cancers de l’Ovaire (GINECO).” Cancer 115 (6): 1234–44. https://doi.org/10.1002/cncr.24149.
Chai, Edna Zhi Pei, Muthu K. Shanmugam, Frank Arfuso, Arunasalam Dharmarajan, Chao Wang, Alan Prem Kumar, Ramar Perumal Samy, et al. 2016. “Targeting Transcription Factor STAT3 for Cancer Prevention and Therapy.” Pharmacology & Therapeutics 162 (June): 86–97. https://doi.org/10.1016/j.pharmthera.2015.10.004.
Cooper, M. A., T. A. Fehniger, and M. A. Caligiuri. 2001. “The Biology of Human Natural Killer-Cell Subsets.” Trends in Immunology 22 (11): 633–40. https://doi.org/10.1016/s1471-4906(01)02060-9.
Davies, Richard, Petra Vogelsang, Roland Jonsson, and Silke Appel. 2016. “An Optimized Multiplex Flow Cytometry Protocol for the Analysis of Intracellular Signaling in Peripheral Blood Mononuclear Cells.” Journal of Immunological Methods 436 (September): 58–63. https://doi.org/10.1016/j.jim.2016.06.007.
Fagotti, Anna, Maria Gabriella Ferrandina, Giuseppe Vizzielli, Tina Pasciuto, Francesco Fanfani, Valerio Gallotta, Pasquale Alessandro Margariti, et al. 2020. “Randomized Trial of Primary Debulking Surgery versus Neoadjuvant Chemotherapy for Advanced Epithelial Ovarian Cancer (SCORPION-NCT01461850).” International Journal of Gynecological Cancer: Official Journal of the International Gynecological Cancer Society 30 (11): 1657–64. https://doi.org/10.1136/ijgc-2020-001640.
Guo, Yanwei, Hongqiao Zhang, Xin Xing, Lijuan Wang, Jian Zhang, Lin Yan, Xiaoke Zheng, and Mingzhi Zhang. 2018. “Gelsolin Regulates Proliferation, Apoptosis and Invasion in Natural Killer/T-Cell Lymphoma Cells.” Biology Open 7 (1). https://doi.org/10.1242/bio.027557.
Kang, Chiao-Wen, Avijit Dutta, Li-Yuan Chang, Jayashri Mahalingam, Yung-Chang Lin, Jy-Ming Chiang, Chen-Yu Hsu, et al. 2015. “Apoptosis of Tumor Infiltrating Effector TIM-3+CD8+ T Cells in Colon Cancer.” Scientific Reports 5 (October): 15659. https://doi.org/10.1038/srep15659.
Kuroki, Lindsay, and Saketh R. Guntupalli. 2020. “Treatment of Epithelial Ovarian Cancer.” BMJ 371 (November): m3773. https://doi.org/10.1136/bmj.m3773.
Onuma, Toshimichi, Meshach Asare-Werehene, Yoshio Yoshida, and Benjamin K. Tsang. 2022. “Exosomal Plasma Gelsolin Is an Immunosuppressive Mediator in the Ovarian Tumor Microenvironment and a Determinant of Chemoresistance.” Cells 11 (20). https://doi.org/10.3390/cells11203305.
Wang, Xu, Wenbin Xin, Hua Zhang, Fengmei Zhang, Meilan Gao, Lingling Yuan, Xiaoyan Xu, Xuemei Hu, and Mingdong Zhao. 2014. “Aberrant Expression of P-STAT3 in Peripheral Blood CD4+ and CD8+ T Cells Related to Hepatocellular Carcinoma Development.” Molecular Medicine Reports 10 (5): 2649–56. https://doi.org/10.3892/mmr.2014.2510.
Reviewer 3 Report
Comments and Suggestions for Authors
This study aims to investigate the effects of increased pGSN expression by ovarian cancer cells.In chemoresistant conditions, ovarian cancer cells (OVCA) with increased pGSN expression exhibit higher levels of both pGSN and TIM-3, leading to increased apoptosis and decreased IFNγ production in activated NK cells. Conversely, pGSN levels are relatively lower in chemosensitive conditions, resulting in an inability to suppress the function of activated NK cells. This article provides fresh insights into the immunological aspects of chemoresistant ovarian cancer. However, there are a couple of issues to address:
- Figure 1C does not convincingly demonstrate that some activated NK cells were positive for pGSN expression, which may be attributed to uptake given the typically low levels of pGSN in NK cells, which are important for their motility and actin remodeling.
- The statement about pGSN levels in the conditioned medium (CM) from A2780S cells with pGSN overexpression being increased compared to A2780S without pGSN overexpression is incorrectly referenced as Supplementary Figure S1D; it should be Supplementary Figure 2D.
- The effects of pGSN overexpression and siRNA knockdown in TOV3041G cells are not significant.
- Baseline levels of pGSN expression and secretion in chemosensitive (TOV3041G and A2780S) versus chemoresistant (OV90 and A2780CP) cells need further clarification.
- Are there changes in other chemokines or cytokines, besides IFNγ, due to altered pGSN levels?
- In Figure 2A, it is shown that CM from both chemosensitive (TOV3041G and A2780S) and chemoresistant (OV90 and A2780CP) cells can decrease NK cell proliferation. The reasons for this observation should be explored further.
- What are the potential mechanisms by which pGSN induces apoptosis in NK cells?
Author Response
Dear Reviewers,
We would like to express our deepest gratitude for your invaluable assistance during the review process of our manuscript. Your thoughtful comments and insightful suggestions have significantly contributed to the improvement of our work.
Your thorough evaluation and constructive feedback have helped us refine the clarity and coherence of our research. We sincerely appreciate the time and expertise you have generously shared with us, as your input has been instrumental in strengthening the overall quality of our manuscript.
We have significantly improved the manuscript to reflect the new changes and suggestions provided as highlighted below:
Reviewer 3:
This study aims to investigate the effects of increased pGSN expression by ovarian cancer cells.In chemoresistant conditions, ovarian cancer cells (OVCA) with increased pGSN expression exhibit higher levels of both pGSN and TIM-3, leading to increased apoptosis and decreased IFNγ production in activated NK cells. Conversely, pGSN levels are relatively lower in chemosensitive conditions, resulting in an inability to suppress the function of activated NK cells. This article provides fresh insights into the immunological aspects of chemoresistant ovarian cancer. However, there are a couple of issues to address:
- Figure 1C does not convincingly demonstrate that some activated NK cells were positive for pGSN expression, which may be attributed to uptake given the typically low levels of pGSN in NK cells, which are important for their motility and actin remodeling.
Response: As rightly pointed out by the reviewer, pGSN uptake in the tissues will be difficult to prove. However, expression of pGSN in NK cells is very low. The cell line experiments suggest that pGSN from ovarian cancer are transmitted to NK cells as demonstrated by our flow cytometry results. We added the images of pGSN negative NK cell in Supplementary Figure S3B. This study did not demonstrate that pGSN is involved in motility or actin remodeling with respect to NK cells, so this sentence has been removed. Line 296: “Some of the activated NK cells were negative for pGSN (Supplementary Figure S3B). Interestingly, some of the activated NK cells were positive for pGSN expression, which could be due to uptake since NK cells have low levels of pGSN [20] (Supplementary Figure S1B and Figure 1C).
- The statement about pGSN levels in the conditioned medium (CM) from A2780S cells with pGSN overexpression being increased compared to A2780S without pGSN overexpression is incorrectly referenced as Supplementary Figure S1D; it should be Supplementary Figure 2D.
Response: Thank you very much for pointing this out. We have made the corrections. The following revisions have been made: Line 216, “NK92MI cells were treated with 25 ng/ml PMA and 1 μg/ml ionomycin and rhpGSN or conditioned media (CM) from A2780S cells overexpressing pGSN (Supplementary Figure 2D).”
- The effects of pGSN overexpression and siRNA knockdown in TOV3041G cells are not significant.
Response: A small increase in pGSN overexpression was observed in WB for TOV3041G. TOV3041G cells grew slowly, which may have been a contributing factor. Although the change in pGSN overexpression in TOV3041G was small, pGSN overexpression increased NK cell apoptosis and TIM-3 expression. We believe that this indicates a strong immunosuppressive effect of pGSN.
- Baseline levels of pGSN expression and secretion in chemosensitive (TOV3041G and A2780S) versus chemoresistant (OV90 and A2780CP) cells need further clarification.
Responses: Our previous studies have demonstrated baseline expressions and secretions of pGSN in OVCA chemoresistant and sensitive cells (Asare-Werehene, Communal, et al. 2020; Asare-Werehene, Nakka, et al. 2020). We have shown that chemoresistant OVCA cells (OV90 and A2780cp) express higher levels of pGSN compared with the sensitive cells (A2780s and TOV3041G) (Asare-Werehene, Communal, et al. 2020). Thus, our current study is an extension of our previous findings.
- Are there changes in other chemokines or cytokines, besides IFNγ, due to altered pGSN levels?
Response: We did not assess other cytokine profile in the current study. We have previously shown that pGSN causes apoptosis in CD8+ T cells and decrease IFNγ expression as well as increases the expression of IL-4 in CD4+ T cells. (Asare-Werehene, Communal, et al. 2020). Interestingly, pGSN itself also acts as a chemokine which selectively attracts and suppresses the viability of M1 macrophages without affecting M2 macrophages (Asare-Werehene et al. 2022). This suggests that in patients with high pGSN expression, M1 macrophages are selectively attracted into the cancer islet, and their viability is reduced without affecting the viability of M2 macrophages. Thus, the M1/M2 ratio is significantly decreased which is associated with poor survival and chemoresistance (Asare-Werehene et al. 2022).
- In Figure 2A, it is shown that CM from both chemosensitive (TOV3041G and A2780S) and chemoresistant (OV90 and A2780CP) cells can decrease NK cell proliferation. The reasons for this observation should be explored further.
Response: Previous reports have shown that overexpression of pGSN on NK cells inhibits AKT-PI3K axis and suppresses cell proliferation (Guo et al. 2018). Conditioned media derived from cancer cells contain various factors that suppress immune cell function. Chemoresistant cells contain higher levels of pGSN. According to Figure 2B and 2C, pGSN might be transmitted to NK cells and suppressed cell proliferation. The following revisions have been made: Line 568, “Exogenous recombinant human pGSN and OVCA-derived CM enriched in pGSN could suppress NK cell proliferation. Previous reports have shown that overexpression of pGSN on NK cells inhibits AKT-PI3K axis and suppresses cell proliferation [14]. Exogenous recombinant human pGSN and OVCA-derived CM enriched in pGSN might be transmitted to NK cell and suppress cell proliferation.”
- What are the potential mechanisms by which pGSN induces apoptosis in NK cells?
Response: pGSN induces caspase-3-dependent cell death in CD8+ T cells and M1 macrophages (Asare-Werehene, Communal, et al. 2020; Asare-Werehene et al. 2022). However, in our current study, significant apoptosis was only observed during cell-to-cell contact, suggesting that TIM-3 and galectin-9 might be potentially involved. Also, TIM-3 positive CD8+ T cells exhibited more apoptosis than the TIM-3 negative CD8+ T cells (Kang et al. 2015) and Galectin-9/TIM-3 signaling blockade with anti-TIM-3 antibody reduced the apoptosis. Similarly, Galectin-9/TIM-3 blockade results in tumor growth inhibition in mice (Kang et al. 2015). We therefore hypothesize that increased pGSN induces TIM-3 over-expression in NK cells resulting in the initiation of pro-apoptotic signaling pathways. The following revisions have been made: Line 581, “TIM-3 and galectin-9 might be deeply involved in NK cell apoptosis by pGSN. The TIM-3 positive CD8+ T cells exhibited more apoptosis than the TIM-3 negative CD8+ T cells [55]. Galectin-9/TIM-3 signaling blockade with anti-TIM-3 antibody reduces the apoptosis and in addition, inhibits tumor growth in mice [55]. Blocking pGSN synthesis as well as the interaction of TIM-3 and gelactin-9 could enhance NK cell activity and function, ultimately reducing apoptosis. This could help effectively kill OVCA cells.”
Reference:
Abedini, Mohammad R., Pei-Wen Wang, Yu-Fang Huang, Mingju Cao, Cheng-Yang Chou, Dar-Bin Shieh, and Benjamin K. Tsang. 2014. “Cell Fate Regulation by Gelsolin in Human Gynecologic Cancers.” Proceedings of the National Academy of Sciences of the United States of America 111 (40): 14442–47. https://doi.org/10.1073/pnas.1401166111.
Asare-Werehene, Meshach, Laudine Communal, Euridice Carmona, Youngjin Han, Yong Sang Song, Dylan Burger, Anne-Marie Mes-Masson, and Benjamin K. Tsang. 2020. “Plasma Gelsolin Inhibits CD8+ T-Cell Function and Regulates Glutathione Production to Confer Chemoresistance in Ovarian Cancer.” Cancer Research 80 (18): 3959–71. https://doi.org/10.1158/0008-5472.CAN-20-0788.
Asare-Werehene, Meshach, Laudine Communal, Euridice Carmona, Tien Le, Diane Provencher, Anne-Marie Mes-Masson, and Benjamin K. Tsang. 2019. “Pre-Operative Circulating Plasma Gelsolin Predicts Residual Disease and Detects Early Stage Ovarian Cancer.” Scientific Reports 9 (1): 13924. https://doi.org/10.1038/s41598-019-50436-1.
Asare-Werehene, Meshach, Kiran Nakka, Arkadiy Reunov, Chen-Tzu Chiu, Wei-Ting Lee, Mohammad R. Abedini, Pei-Wen Wang, et al. 2020. “The Exosome-Mediated Autocrine and Paracrine Actions of Plasma Gelsolin in Ovarian Cancer Chemoresistance.” Oncogene 39 (7): 1600–1616. https://doi.org/10.1038/s41388-019-1087-9.
Asare-Werehene, Meshach, Hideaki Tsuyoshi, Huilin Zhang, Reza Salehi, Chia-Yu Chang, Euridice Carmona, Clifford L. Librach, et al. 2022. “Plasma Gelsolin Confers Chemoresistance in Ovarian Cancer by Resetting the Relative Abundance and Function of Macrophage Subtypes.” Cancers 14 (4). https://doi.org/10.3390/cancers14041039.
Bois, Andreas du, Alexander Reuss, Eric Pujade-Lauraine, Philipp Harter, Isabelle Ray-Coquard, and Jacobus Pfisterer. 2009. “Role of Surgical Outcome as Prognostic Factor in Advanced Epithelial Ovarian Cancer: A Combined Exploratory Analysis of 3 Prospectively Randomized Phase 3 Multicenter Trials: By the Arbeitsgemeinschaft Gynaekologische Onkologie Studiengruppe Ovarialkarzinom (AGO-OVAR) and the Groupe d’Investigateurs Nationaux Pour Les Etudes Des Cancers de l’Ovaire (GINECO).” Cancer 115 (6): 1234–44. https://doi.org/10.1002/cncr.24149.
Chai, Edna Zhi Pei, Muthu K. Shanmugam, Frank Arfuso, Arunasalam Dharmarajan, Chao Wang, Alan Prem Kumar, Ramar Perumal Samy, et al. 2016. “Targeting Transcription Factor STAT3 for Cancer Prevention and Therapy.” Pharmacology & Therapeutics 162 (June): 86–97. https://doi.org/10.1016/j.pharmthera.2015.10.004.
Cooper, M. A., T. A. Fehniger, and M. A. Caligiuri. 2001. “The Biology of Human Natural Killer-Cell Subsets.” Trends in Immunology 22 (11): 633–40. https://doi.org/10.1016/s1471-4906(01)02060-9.
Davies, Richard, Petra Vogelsang, Roland Jonsson, and Silke Appel. 2016. “An Optimized Multiplex Flow Cytometry Protocol for the Analysis of Intracellular Signaling in Peripheral Blood Mononuclear Cells.” Journal of Immunological Methods 436 (September): 58–63. https://doi.org/10.1016/j.jim.2016.06.007.
Fagotti, Anna, Maria Gabriella Ferrandina, Giuseppe Vizzielli, Tina Pasciuto, Francesco Fanfani, Valerio Gallotta, Pasquale Alessandro Margariti, et al. 2020. “Randomized Trial of Primary Debulking Surgery versus Neoadjuvant Chemotherapy for Advanced Epithelial Ovarian Cancer (SCORPION-NCT01461850).” International Journal of Gynecological Cancer: Official Journal of the International Gynecological Cancer Society 30 (11): 1657–64. https://doi.org/10.1136/ijgc-2020-001640.
Guo, Yanwei, Hongqiao Zhang, Xin Xing, Lijuan Wang, Jian Zhang, Lin Yan, Xiaoke Zheng, and Mingzhi Zhang. 2018. “Gelsolin Regulates Proliferation, Apoptosis and Invasion in Natural Killer/T-Cell Lymphoma Cells.” Biology Open 7 (1). https://doi.org/10.1242/bio.027557.
Kang, Chiao-Wen, Avijit Dutta, Li-Yuan Chang, Jayashri Mahalingam, Yung-Chang Lin, Jy-Ming Chiang, Chen-Yu Hsu, et al. 2015. “Apoptosis of Tumor Infiltrating Effector TIM-3+CD8+ T Cells in Colon Cancer.” Scientific Reports 5 (October): 15659. https://doi.org/10.1038/srep15659.
Kuroki, Lindsay, and Saketh R. Guntupalli. 2020. “Treatment of Epithelial Ovarian Cancer.” BMJ 371 (November): m3773. https://doi.org/10.1136/bmj.m3773.
Onuma, Toshimichi, Meshach Asare-Werehene, Yoshio Yoshida, and Benjamin K. Tsang. 2022. “Exosomal Plasma Gelsolin Is an Immunosuppressive Mediator in the Ovarian Tumor Microenvironment and a Determinant of Chemoresistance.” Cells 11 (20). https://doi.org/10.3390/cells11203305.
Wang, Xu, Wenbin Xin, Hua Zhang, Fengmei Zhang, Meilan Gao, Lingling Yuan, Xiaoyan Xu, Xuemei Hu, and Mingdong Zhao. 2014. “Aberrant Expression of P-STAT3 in Peripheral Blood CD4+ and CD8+ T Cells Related to Hepatocellular Carcinoma Development.” Molecular Medicine Reports 10 (5): 2649–56. https://doi.org/10.3892/mmr.2014.2510.